# A non-B DNA binding peptidomimetic channel alters cellular functions

Raj Paul [1], Debasish Dutta[1], Titas Kumar Mukhopadhyay[1], Diana Müller[2], Binayak Lala[1], Ayan Datta [1], Harald Schwalbe [2] & Jyotirmayee Dash [1] ✉

DNA binding transcription factors possess the ability to interact with lipid membranes to construct ion-permeable pathways. Herein, we present a thiazole-based DNA binding peptide mimic **TBP2**, which forms transmembrane ion channels, impacting cellular ion concentration and consequently stabilizing G-quadruplex DNA structures. **TBP2** self-assembles into nanostructures, e.g., vesicles and nanofibers and facilitates the transportation of $Na^+$ and $K^+$ across lipid membranes with high conductance (~0.6 nS). Moreover, **TBP2** exhibits increased fluorescence when incorporated into the membrane or in cellular nuclei. Monomeric **TBP2** can enter the lipid membrane and localize to the nuclei of cancer cells. The coordinated process of time-dependent membrane or nuclear localization of **TBP2**, combined with elevated intracellular cation levels and direct G-quadruplex (G4) interaction, synergistically promotes formation and stability of G4 structures, triggering cancer cell death. This study introduces a platform to mimic and control intricate biological functions, leading to the discovery of innovative therapeutic approaches.

Bio-inspired peptides exhibit a remarkable ability for self-assembly into nanostructures in specific microenvironments, enabling the formation of membrane channels upon insertion into biological or model lipid bilayers[1,2]. Some of these peptides induce cytotoxicity or potent anti-tumor activity[2]. Transmembrane channel proteins play a vital role in cellular homeostasis by transporting ions across lipid bilayer membranes[1–3]. However, their practical utility is limited by structural complexity and critical molecular mechanisms[4–6]. Efforts have been directed towards the development of synthetic ion channels with nucleic acids, natural transcription factors and peptides, and synthetic molecules by mimicking characteristics of biological ion channels[7–16]. Only a few small molecule ion transporters have been reported to exhibit therapeutic properties[17–23]. For instance, Zhang et al. reported a small molecule based cation transporter with the ability to kill cancer cells[19]. Crafting synthetic molecular ion transporters with therapeutic potential and understanding their biophysical and biological properties could provide critical insights into their functional mechanisms[24–36].

Peptide mimetics[37–41], due to their unique structural diversity and potential role in biological systems, hold promise as essential chemotherapeutic agents by targeting DNA or its secondary structures.

Non-B-DNA four stranded secondary structures e.g., G-quadruplexes (G4s) play a crucial role in cellular growth by regulating the replication and transcription machinery[42–44]. G4s are commonly found in the promoter G-rich region of various proto-oncogenes (e.g., *c-MYC*, *c-KIT*), as well as in telomeres (*h-TELO*) of cancer cells. The *c-MYC* proto-oncogene predominantly controls cell proliferation, apoptosis, and drug resistance in various cancer types such as cervical, breast, and lung cancers[44–48]. The NHE (nuclease hypersensitive element) $III_1$ region of *c-MYC* promoter is responsible for approximately 90% of its transcriptional activity, contains G4-forming sequences that act as transcriptional repressors[45–52]. Stabilizing these G4 structures with synthetic molecules has emerged as a potential strategy for cancer therapeutics, offering promising avenues for targeted cancer treatment[53–59].

[1]School of Chemical Sciences, Indian Association for the Cultivation of Science, Kolkata 700032, India. [2]Institute of Organic Chemistry and Chemical Biology, Center for Biomolecular Magnetic Resonance (BMRZ), Goethe, University Frankfurt, Max-von-Laue Strasse 7, 60438 Frankfurt am Main, Germany. ✉ e-mail: ocjd@iacs.res.in

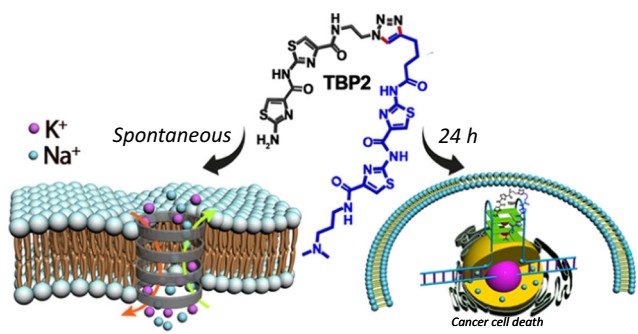

**Fig. 1 | Time-dependent functions of peptidomimetic channel. TBP2** forming gated ion channels and inducing cell death.

In this work, we show the development of a thiazole-based dimeric peptidomimetic (**TBP2**), which exhibits the unique ability to form self-assembled nanostructures in lipid microenvironments and effectively transport ions across model lipid bilayer membranes (Fig. 1). Moreover, **TBP2** inhibits cancer cell growth by modulating intracellular ion concentrations and G4-mediated transcriptional regulation of oncogenes. This study represents the illustration of a DNA binding artificial ion channel, introducing an approach for engineering self-assembling peptidomimetics that can mimic and regulate essential biological functions (Fig. 1).

## Results and discussion

### Design and synthesis of self-assembling peptidomimetics

Thiazole peptides (**TBP**) were synthesized by the cycloaddition of azide and alkyne functionalized bis-thiazoles **1** and **2** in the presence of copper sulfate and sodium ascorbate in a solvent mixture of *tert*-butanol/water (Fig. 2(i), Fig. S1 and details in Supplementary Information). Bis-thiazole azide **1** and alkyne **2** were prepared using stepwise amide coupling, resulting in high overall yields. The dimeric peptide **TBP2** was obtained by Boc-deprotection of **TBP1**. Due to the presence of dimeric bis-thiazole amide units connected via a triazole linkage, these peptidomimetics were expected to self-assemble into supramolecular nanoarchitectures, capable of transitioning between open and folded conformations in response to an external stimulus (Fig. 2i). Moreover, the thiazole units and aliphatic amino group were anticipated to facilitate the permeation of **TBP**s into cellular nuclei and target DNA secondary structures via stacking or groove binding mode[39,40].

### Supramolecular self-assembly of TBPs into vesicles and nanofibres

Transmission Electron Microscopy (TEM) imaging revealed that **TBP2** could self-organize into nanofibrous and vesicle like structures in both NaCl and KCl buffers. The dynamic light scattering (DLS) study further revealed the relative abundance of vesicles formed through the supramolecular assembly of the peptidomimetic **TBP2**, showing an average size distribution of approximately 10 nm (Fig. 2(ii)B). The formation of multiple vesicular structures could presumably be due to both the hydrophobic and hydrophilic parts of the compounds and strong aggregation of aromatic surfaces.

However, in the case of peptidomimetic **TBP1** with the Boc protecting group, no ordered structures similar to those of **TBP2** were observed in the presence of either Na$^+$ or K$^+$ ions (see the Supplementary Information, Fig. S2). In the TEM image of **TBP1**, only a limited number of vesicular structures were observed, and they were not as distinct as those seen in **TBP2**. Consequently, the presence of the -NH$_2$ group in **TBP2** played a crucial role in promoting the formation of well-organized non-covalent structures. The supramolecular structure of **TBP2** was determined by optimizing the molecule using quantum

chemistry calculations at different levels (M06-2x/6-31 G(d,p) and B3LYP/6-31 G(d,p)). The optimized structure revealed that **TBP2** became folded and stabilized by three hydrogen bonding interactions, one of which involves the terminal -NH$_2$ group (Fig. 2(ii)D, (ii)E). The hydrogen bonding distances were calculated to be 1.85 Å, 1.88 Å, and 2.31 Å, respectively.

The structure exhibited an almost elliptical cavity, with dimensions of approximately 5.8 Å for the long axis and 4.6 Å for the short axis. Subsequently, Na$^+$/K$^+$ ions were placed within this cavity, and their coordination with the molecular structure was investigated through DFT optimization (Fig. 2(ii)F, (ii)G). The cations were observed to be well-contained within the cavity without causing significant changes to the surrounding molecular arrangement. The cations were coordinated to three donor atoms, a triazole nitrogen, a nitrogen atom from a thiazole ring and a carbonyl oxygen from a peptide linkage, and the corresponding distances are shown in Fig. 2(ii) F, (ii) G.

To quantify ion encapsulation, the binding energies of Na$^+$ and K$^+$ were calculated using the formula $E_{binding} = E_{(M-ion)}-E_M-E_{ion}$, where $E_{(M-ion)}$, $E_M$, and $E_{ion}$ represent the electronic energies of the optimized structures of the molecule-ion composite system, the free molecule, and the cation, respectively. The binding energies for Na$^+$ and K$^+$ were calculated to be −75.6 kcal/mol and −55.9 kcal/mol, respectively at the M06-2x/6-31 G(d,p), and −74.6 kcal/mol, and −67.7 kcal/mol, respectively at B3LYP/6-31 G(d,p) level. The high binding energies indicate the strong stability of the folded molecular structure to accommodate Na$^+$ and K$^+$ ions within its cavity, and are similar to those found for these metal ions bound to the active sites of various proteins[60]. We further optimized the structure of two such folded molecules on top of each other, following periodic propagation to create a configuration which contains six vertically stacked molecules. Molecular dynamics energy minimization was performed for geometry relaxation, resulting in a structure where the vertical stacking of **TBP2** maintained the pore arrangement observed in a single **TBP2** molecule (Fig. 2(ii)C). The top view of this stacked arrangement confirmed the preservation of the pore structure (see Supplementary Information, Fig. S9d).

### TBP2 transports cations across model membrane bilayers

Fluorescence based HPTS (8-hydroxypyrene-1, 3, 6-trisulfonic acid trisodium salt) assay was employed[27] to investigate the potential formation of supra-molecular channel-like nanostructures and ion transport activity across lipid membranes (Fig. 3). HPTS was encapsulated in large unilamellar vesicles (LUVs) of average diameter ~70 nm (dynamic light scattering method), prepared from egg yolk L-α-phosphatidylcholine (EYPC) in HEPES NaCl or HEPES KCl buffer (10 mM HEPES, 100 mM NaCl or KCl, pH 6.4).

The liposomes were then suspended in 10 mM HEPES buffer (pH 6.4) containing 100 mM of MCl$_2$ (Mn$^+$ = Na$^+$, K$^+$, Cs$^+$, Rb$^+$, Li$^+$). After incorporating **TBP1** and **TBP2** (dissolved in DMSO) into the bilayer membrane, an external pH gradient was generated by adding NaOH. The change in HPTS fluorescence intensity was recorded over time. **TBP1** exhibited minimal changes in HPTS fluorescence up to 20 μM, while **TBP2** displayed high transport activity for K$^+$ (~70%) and Na$^+$ (~65%) in the presence of HEPES NaCl or HEPES KCl internal buffer (pH 7.4) (Fig. 3).

Using the Hill equation, EC$_{50}$ (concentration of molecules to achieve 50% ion transport activity) values were determined to be 3.1 μM and 14 μM for **TBP2**, whereas **TBP1** showed EC$_{50}$ values of 43 μM and 56 μM for K$^+$ and Na$^+$, respectively (Internal buffer: HEPES NaCl, pH 6.4) (Fig. 2 and see the Supplementary Information, Fig. S3). Substituting the internal buffer with KCl (10 mM HEPES, 100 mM KCl, pH 6.4) resulted in **TBP2** displaying EC$_{50}$ values of 5.8 μM and 7 μM for K$^+$ and Na$^+$, respectively. **TBP2** exhibited relatively greater transport efficiency for K$^+$ and Na$^+$ in the presence of alternate internal buffers, possibly due to the exchange of K$^+$ and Na$^+$. The Hill coefficient of **TBP2** for K$^+$ (nK$^+$ = 1.64) indicates that more than one molecule (i.e., positive

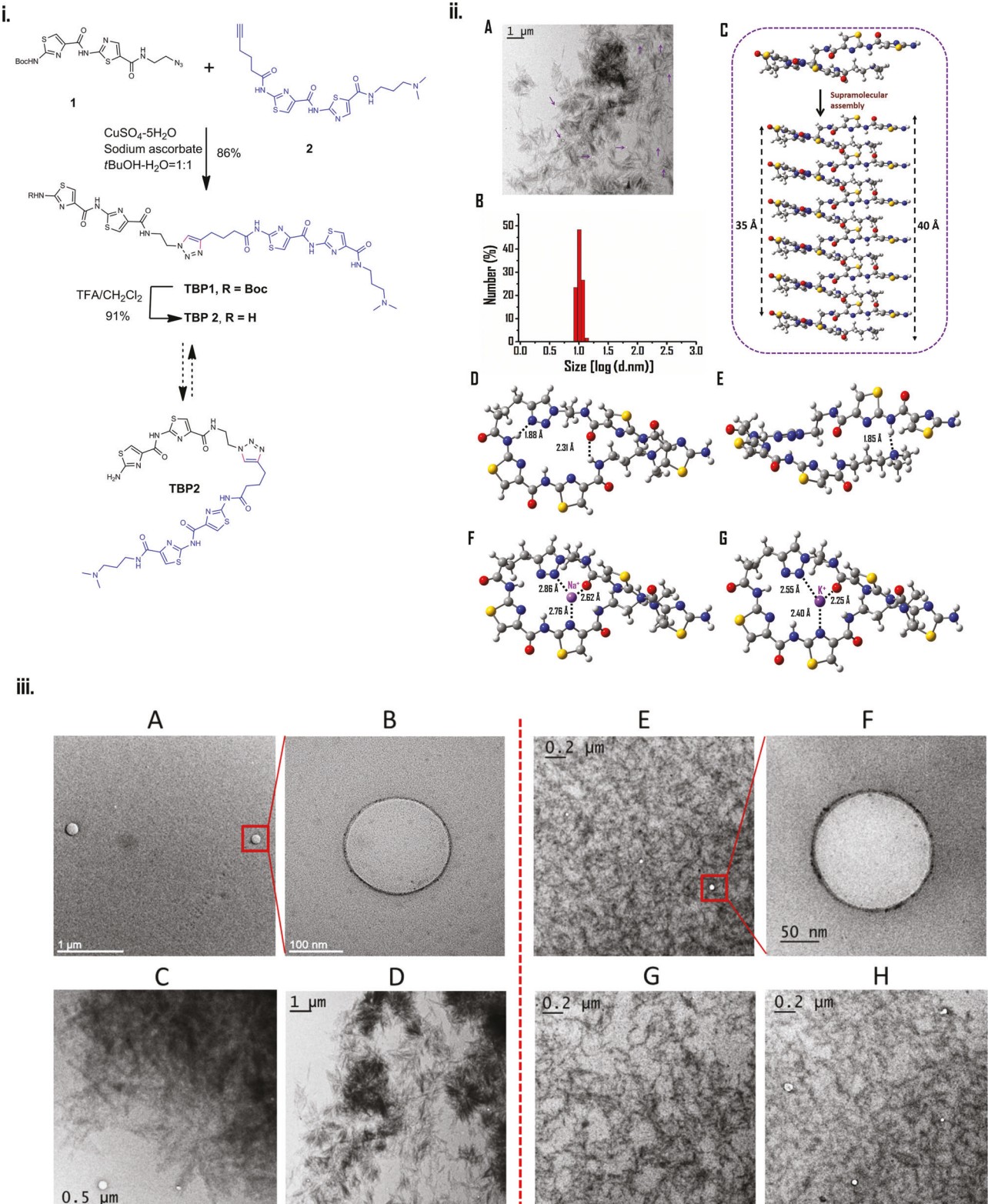

**Fig. 2 | Supramolecular self-assembly of TBP2.** (i) Design and synthesis of dimeric thiazole peptidomimetics. (ii) Self-assembling properties of **TBP2**: (**A**) TEM micrograph of **TBP2** showing distribution of vesicles (indicated with purple arrow). The TEM imaging were performed independently three times ($n = 3$). **B** The average size distribution (±SD) and abundance of **TBP2** formed vesicles as measured by DLS method. **C** Vertically stacked supramolecular arrangement of six **TBP2** molecules forming the ion channel, as obtained from the periodic propagation of an optimized **TBP2** dimer followed by molecular dynamics minimization. **D** Top and (**E**) side views of the optimized structure of a single **TBP2** molecule at the M06-2x/6-31 G(d,p) level of theory. The hydrogen bond distances are indicated in the Figure (**F**, **G**) Optimized structure of Na⁺ and K⁺ encapsulated **TBP2**, respectively. Each of the cations is coordinated with three donor atoms and the corresponding distances are indicated in the Figure. (iii) High Resolution TEM micrographs of **TBP2** in NaCl (**A**–**D**) and KCl (**E**–**H**) buffers (pH 7.4); showing nanofibre and vesicular structures. The experiments were performed independently at least three times ($n = 3$).

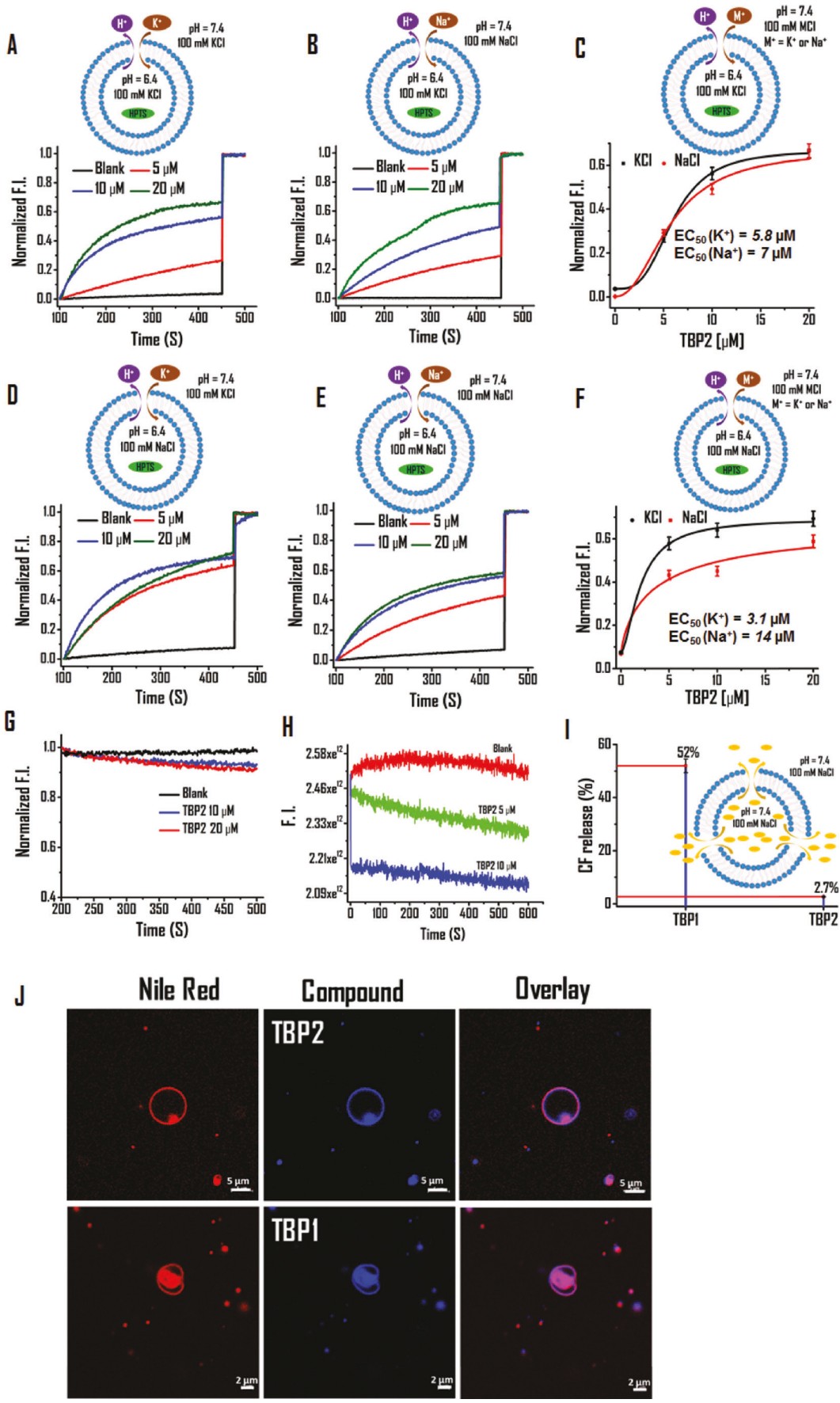

**Fig. 3 | TBP2 gated ion transport via model lipid bilayer. A–F** HPTS assay for measuring the ion transport activity of **TBP2** in the presence of Na$^+$ and K$^+$ ions. Buffer composition: Internal – (**A, B**) 10 mM HEPES 100 mM KCl (pH 6.4), (**D, E**) 10 mM HEPES 100 mM NaCl (pH 6.4), External – (**A, E**) 10 mM HEPES 100 mM KCl (pH 7.4), (**B, D**) 10 mM HEPES 100 mM NaCl (pH 7.4). **C** EC$_{50}$ value determination of **TBP2** in the presence of external buffer – 10 mM HEPES 100 mM NaCl or KCl (pH 7.4), internal buffer – 10 mM HEPES 100 mM KCl (pH 6.4). Data are presented as means ± SD (*n* = 3). Source data are available. **F** EC$_{50}$ value determination of **TBP2** in the presence of external buffer – 10 mM HEPES 100 mM NaCl or KCl (pH 7.4), internal buffer – 10 mM HEPES 100 mM NaCl (pH 6.4). Data are represented as means ± SD (*n* = 3). Source data are available. **G** Lucigenin assay of **TBP2** for Cl$^-$

transport. Change in fluorescence intensity as a function of time in 225 mM NaNO$_3$ buffer. **H** Safranin O assay for membrane polarization in the presence of HEPES-NaCl (external) and HEPES-KCl (internal) buffers. **I** CF release assay; determination of CF release percentage in the presence of **TBP1** and **TBP2** (External: 10 mM HEPES 100 mM NaCl, pH 7.4; Internal: 10 mM HEPES 100 mM NaCl, pH 7.4) after 8 minutes. Error bars represent ± SEM (*n* = 3) (**J**) Membrane colocalization of **TBP1** and **TBP2** in GUVs. Scale bars represent 5 μM (top row) and 2 μM (bottom row). The imaging experiments were performed independently at least three times (*n* = 3). HPTS, 8-hydroxypyrene-1, 3, 6-trisulfonic acid trisodium salt; CF Carboxyfluorescein, LUVs large unilamellar vesicles, GUVs giant unilamellar vesicles.

co-operativity) might form supramolecular ion channels to transport K$^+$. Thus, HPTS studies revealed that **TBP2** could form channel like structures, facilitating efficient transportation of K$^+$ and Na$^+$, whereas **TBP1** did not show channel-forming activities. The HPTS data further revealed EC$_{50}$ values of 54 ± 2.7 μM, 26 ± 1.3 μM, and 14 ± 0.7 μM for Li$^+$, Cs$^+$, and Rb$^+$, respectively, indicating greater transport efficiency of **TBP2** for K$^+$ (EC$_{50}$ = 5.8 ± 0.3 μM) and Na$^+$ (EC$_{50}$ = 7 ± 0.4 μM) (see the Supplementary Information, Fig. S3.2). Based on HPTS data, the transport efficiency of **TBP2** for different alkali cations can be inferred as K$^+$ > Na$^+$ > Rb$^+$ > Cs$^+$ » Li$^+$.

Furthermore, we conducted a lucigenin assay to observe Cl$^-$ transport activity of **TBP2**, where EYPC-LUVs were filled with the lucigenin dye in 225 mM NaNO$_3$ solution and the fluorescence intensity of the lucigenin ($\lambda_{em}$ = 535 nm; $\lambda_{ex}$ = 455 nm) was monitored until 500 sec (Fig. 3G). No significant changes in lucigenin fluorescence were observed with incremental addition of **TBP2**; indicating no Cl$^-$ flux across the lipid membrane. Altogether, HPTS and lucigenin assay demonstrate that **TBP2** preferentially transports cations over anions across unilamellar vesicles.

## Membrane integrity and polarization evaluation

To assess the impacts of **TBP1** and **TBP2** on membrane integrity, a CF leakage assay was conducted using a self-quenched carboxy-fluorescein dye (CF) (<10 Å in size) (Fig. 3I, see Supplementary Information, Fig. S4). The membrane-impermeable CF dye (40 mM) can efflux from the vesicles upon pore formation (>10 Å) or disruption of the LUVs, leading to an increase in fluorescence intensity. After addition of **TBP1** and **TBP2**, the CF discharge percentage was calculated as 52% and 2.7% after 8 minutes, respectively. The lower CF release in the presence of **TBP2** indicated conserved membrane integrity of LUVs, while the higher CF discharge suggested that **TBP1** could disrupt the membrane structure or create larger openings in the vesicles.

Furthermore, a safranin O assay was performed to investigate **TBP2**-dependent membrane polarization using unilamellar vesicles. Upon addition of safranin O to the vesicular solution, the fluorescence (with or without **TBP2**) (1 μM) was monitored for 600 seconds at an excitation of 522 nm and emission of 581 nm (Fig. 3H). The data revealed a significant decrease in the fluorescence intensity of safranin O with increasing ligand concentration, indicating **TBP2**-induced membrane polarization in the presence of alternate NaCl or KCl buffers. These results demonstrate that **TBP2** embeds within the membrane and forms channels to facilitate ion transport via the vesicular system.

## Membrane localization and conductance measurement

Confocal microscopy was employed to examine the membrane embedding feature of **TBP1** and **TBP2** using Giant Unilamellar Vesicles (GUVs) and HeLa (cervical carcinoma) cells stained with nile red. The microscopic imaging results revealed that **TBP2** could effectively embed within the GUV membrane and co-localize with nile red, the membrane staining dye (Fig. 3J). In contrast, **TBP1** appeared to disrupt the membrane structure, corroborating the findings from the CF

release assay, where **TBP2** preserved the membrane integrity of the vesicles. Further, the imaging data confirmed the membrane insertion properties of the peptidomimetics **TBP1** and **TBP2** in HeLa cells, showing their rapid colocalization with nile red within the cell membrane (see the Supplementary Information, Fig. S5).

To gain real-time insights into the channel-forming behavior of the thiazolyl peptidomimetic **TBP2**, patch clamp experiments were performed using planar lipid bilayers. *Cis* and *trans* compartments, containing 1 M NaCl or 1 M KCl solutions, were separated by a planar lipid bilayer membrane composed of EYPC lipid. Currents were measured over time against different applied potentials (+ve and −ve). Upon adding **TBP2** to the planar bilayer, distinct channel openings and closings were observed at +80 mV and −80 mV, providing the formation of channels across planar lipid bilayer membrane in the presence of both Na$^+$ and K$^+$ (Fig. 4A–D). When 1 M NaCl was present on both the *cis* and *trans* sides, **TBP2** demonstrated multiple channel openings for Na$^+$ transport at −80 mV (Fig. 4A). At the holding potential of +80 mV, it displayed multiple square-top behavior in the presence of either NaCl or KCl (Fig. 4C, D). **TBP2** formed stable channel openings while transporting Na$^+$, whereas its channel openings fluctuated more rapidly between closed and open conformations during K$^+$ transport (Fig. 4). Furthermore, **TBP2** demonstrated continuous channel openings and closings for a longer duration (measured up to -15 seconds) in the presence of 1 M KCl at the applied potential of +100 mV, suggesting its potential to form channel-like structures within the lipid membrane (see the Supplementary Information, Fig. S6). **TBP2** exhibited efficient transport of Na$^+$ and K$^+$ across planar lipid bilayer membrane with high conductance. The I-V plot of **TBP2** exhibited an ohmic-linear relationship between current vs. voltage (Fig. 4G). The average conductance values for transporting Na$^+$ and K$^+$ were measured to be -0.56 nS and -0.68 nS in the presence of 1 M NaCl and KCl (Fig. 4E–J), respectively, indicating robust ion transport capabilities of **TBP2** across the planar lipid bilayer membrane.

The current behavior of **TBP2** was also examined in the presence of CsCl, LiCl, RbCl buffers to explore its voltage-dependent gating characteristics for other monovalent cations (e.g., Cs$^+$, Li$^+$, Rb$^+$) besides Na$^+$ or K$^+$ (see the Supplementary Information, Fig. S7). The I-V analysis revealed that **TBP2** exhibited comparatively lower conductance values for these metal ions (e.g., Cs$^+$: 0.29 nS, Li$^+$: 0.08, Rb$^+$: 0.15 nS), in relation to applied positive or negative voltages. However, no significant currents were observed through **TBP2** incorporated lipid membrane in the absence of applied potentials. These results indicate that **TBP2** possesses distinct preferences for Na$^+$ and K$^+$ ions over other monovalent cations, and its conductance behavior is affected by the specific ions present in the buffer solution.

## Molecular dynamics of ion transport

Molecular dynamics simulation studies were conducted using a vertically stacked arrangement of six **TBP2** molecules embedded within a lipid bilayer of dimensions 10 × 8 nm in the XY plane (Fig. 5A). Subsequently, 500 ns production simulations were initially performed for the ion channel-embedded lipid membrane in 0.15 M NaCl and KCl aqueous solutions, respectively. Throughout the simulations, we

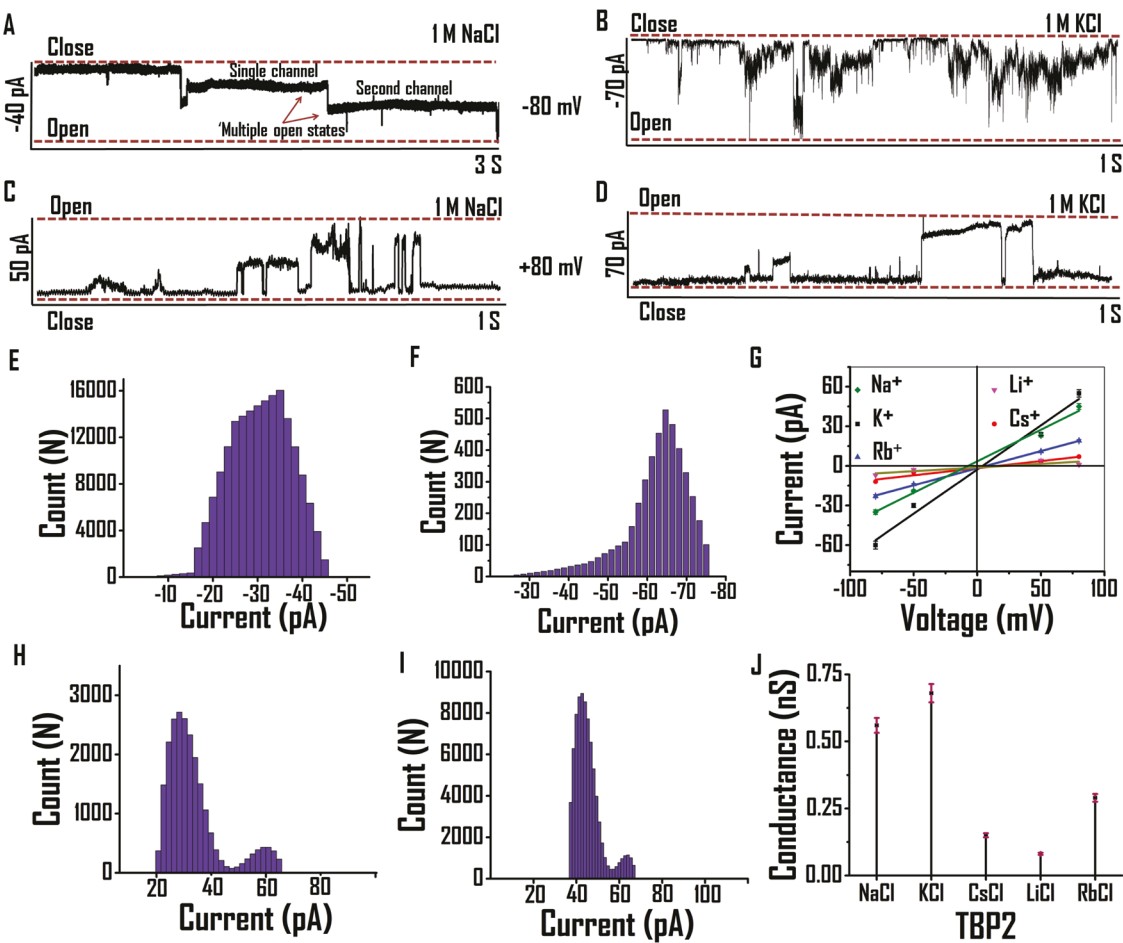

**Fig. 4 | Ionic conductance measurements via patch clamp studies.** Current measurement in the presence of 1 M NaCl at (**A**) −80 mV, (**C**) + 80 mV or 1 M KCl at (**B**) −80 mV and (**D**) + 80 mV, respectively. Bar diagram for frequency vs current of **TBP2** in the presence of 1 M NaCl at (**E**) −80 mV, (**H**) + 80 mV or 1 M KCl at (**F**) −80 mV, (**I**) + 80 mV. **G** I-V plot of **TBP2** in the presence of 1 M MCl [M = Na⁺, K⁺, Cs⁺, Li⁺, Rb⁺] buffers (pH 7). Data are represented as means ± SD ($n$ = 3) (**J**) Drop line plot of conductance of **TBP2** in the presence of different metal ions. Data are represented as means ± SD ($n$ = 3).

monitored the number of $Na^+$ and $K^+$ ions present within the ion channel. Initially, the channel was devoid of any cations. However, within a few picoseconds of simulations (Fig. 5C), cations started entering the channel, and their number fluctuated continuously throughout the entire duration of the simulation.

Most of the time, we observed one or two $Na^+$ and $K^+$ ions inside the channel, with three cations present at fewer time instances. This dynamic variation in the number of cations within the ion channel affirmed the spontaneous passage of ions through the channel, where old ions exited, and new ions entered. The average interaction energy of a single cation with the entire ion channel was calculated to be −23.8 kcal/mol and −14.7 kcal/mol for $Na^+$ and $K^+$, respectively, which aligns with the trends obtained from our quantum chemistry calculations. A sodium ion demonstrated greater stability compared to a potassium ion within the channel. Thus, molecular dynamics simulations shed light on the intricate ion transport dynamics and the preferential stability of $Na^+$ and $K^+$ ions within the **TBP2**-induced ion channel. Additionally, we performed a 2500 ns long simulation with the same system as described above to observe the stability of the assembly of six **TBP2** molecules inside the lipid membrane (Fig. S10, Supplementary Information). The results suggested that **TBP2** was stable inside the lipid membrane throughout the entire duration of MD simulation, Supplementary Movie 1.

To unravel the thermodynamics of ion transport, we calculated the free energy profile for the passage of $Na^+$, $K^+$ and $Cl^-$ through the ion-channel (Fig. 5D). To this regard, we first placed an ion at one end

of the ion channel, and then applied a biasing force, to facilitate its entry and passage through the channel. The resulting free energy profiles exhibit periodic patterns, representing the symmetry in the arrangement of the **TBP2** molecules. The maximum free energy penalty observed for translocation was ~1.8 kcal/mol, ~3 kcal/mol, and ~32 kcal/mol for $Na^+$, $K^+$, and $Cl^-$, respectively. While the magnitudes for the cations were well within the diffusion-controlled limit, favouring their spontaneous passage through the channel; the free energies were significantly high for $Cl^-$, hindering the movement of chloride ions through the channel. Notably, the smaller free energy barriers observed for $Na^+$ can be attributed to its smaller size and greater degree of stabilization within the channel, facilitating its spontaneous passage. On contrary, $K^+$ faced more repulsion to move from the center of one **TBP2** molecule to the next, resulting in relatively sharp free energy barriers for its transport. Given its larger ionic radius and negative charge, chloride experienced pronounced repulsion within the channel, leading to elevated free energy magnitudes. These results suggest that the ion channel had a higher affinity for incorporating smaller $Na^+$ ions, allowing their easier transport compared to $K^+$ ions, while the possibility of $Cl^-$ transport remained weak. The MD simulation study corroborates the findings from HPTS, lucigenin, and patch clamp data, collectively indicating that **TBP2** facilitates the efficient transport of cations like $Na^+$ and $K^+$ with high ionic conductance, while it does not favor the transport of $Cl^-$ ions. Ngo et al. determined the free energies associated with the water-mediated ion transport through a transmembrane

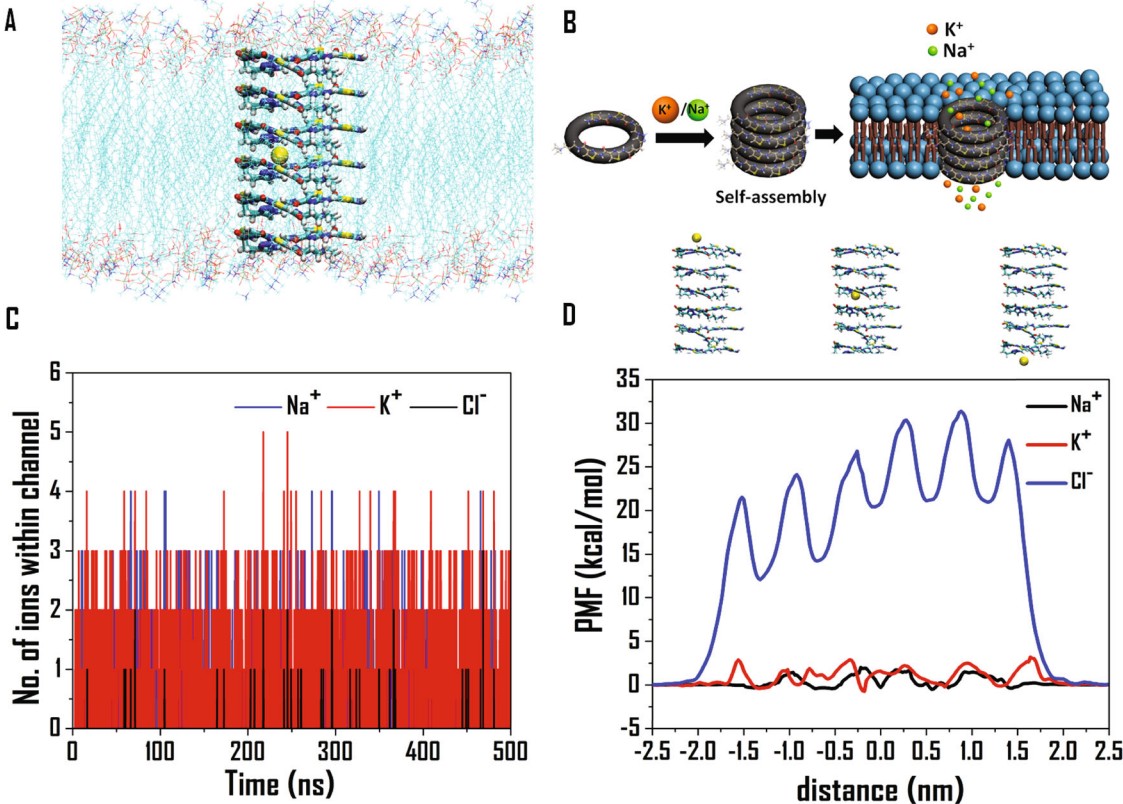

**Fig. 5 | Molecular dynamics simulation of TBP2 gated ion channel. A** Structure of the ion-channel embedded lipid bilayer showing the vertical arrangement of the **TBP2** molecules and Na$^+$ within the channel. Lipid molecules have been blurred for clarity. **B** Schematic illustration of ion channel formation. **C** Number of ions within the ion channel for 500 ns production simulations for the channel-embedded lipid membrane in 0.15 molar NaCl and KCl medium. **D** Free energy profiles (in kcal/mol) in terms of potential of mean forces (PMF's) for the passage of Na$^+$, K$^+$, and Cl$^-$ ions through the ion-channel at 310 K, calculated employing the adaptive biasing force (ABF) module implemented in NAMD 2.12.

ion channel gramicidin A (gA), and found the magnitudes to be 3–5 kcal/mol for both Na$^+$ and K$^{+[60,61]}$. Our results corroborate with their findings, even though on the lower side as compared to those observed for gA.

## TBP2 modulates intracellular Na$^+$ and K$^+$ concentrations in cancer cells

The in-cellulo characteristics of **TBP1** and **TBP2** were first evaluated by monitoring their cellular localization by confocal microscopy. The colocalization study with membrane staining dye nile red revealed that **TBP2** efficiently embedded into the cell membrane immediately after treatment (see the Supplementary Information, Fig. S5). As **TBP2** localized in the cell membrane and exhibited cation transportation (e.g., Na$^+$ and K$^+$) through vesicles, we continued to measure the intracellular concentrations of Na$^+$ and K$^+$ in cancer cells.

To accomplish this, we utilized well-known fluorescent dyes that bind to sodium and potassium, namely sodium-binding benzofuran isophthalate acetoxymethyl ester (SBFI-AM, Na$^+$ probe) and potassium-binding benzofuran isophthalate acetoxymethyl ester (PBFI-AM, K$^+$ probe), respectively[20]. The intracellular fluorescence measurements revealed that **TBP2** (8 μM) significantly[20] increased the concentration of Na$^+$ up to ~ 37% and slightly reduced cytosolic K$^+$ concentration by ~ 15% in HeLa cells (Fig. 6A). Similarly, **TBP2**-embedded A549 cells exhibited a ~19% increase in Na$^+$ concentration and a ~10% decrease in intracellular K$^+$ concentration compared to the control. These results can be rationalized by the different distributions of Na$^+$ and K$^+$ across cell membranes, where intracellular Na$^+$ concentrations (~12 mM) are substantially lower than extracellular Na$^+$ concentrations (~145 mM), and extracellular potassium concentrations

(~4 mM) are significantly lower than its intracellular counterpart (~150 mM) under physiological conditions.

However, after 24 h of incubation in HeLa cells, a significant accumulation of **TBP2** within cellular nuclei was observed via confocal microscopy, possibly due to the molecule's amphipathic nature. Given previous studies demonstrating that thiazole derivatives can bind to non-canonical DNA structures, we investigated the potential of both **TBP1** and **TBP2** to stabilize G4 DNAs in the cellular system. Interestingly, in **TBP2**-treated cells, the number of G4 binding antibody (BG4) foci significantly increased compared to both control cells and cells treated with **TBP1** (Fig. 6B, C). This result suggests that **TBP2** possessed the ability to stabilize G4 structures in cancer cells. Consequently, the data indicated that the significant intracellular increase of monovalent cation Na$^+$ might disrupt the ionic balance, thereby synergistically contributing to the formation and stabilization of G-quadruplex structures. This effect could be a result of substantial accumulation of **TBP2** within the nuclei of cancer cells after an extended incubation period.

The cell growth inhibition assay (see the Supplementary Information, Fig. S11) showed that **TBP2** exhibited IC$_{50}$ values of ~8.2, ~10 and ~16 μM in cervical carcinoma (HeLa), leukemia (K562) and lung carcinoma (A549) cells after 24 h, respectively. Intriguingly, it did not inhibit the growth of normal kidney epithelial (NKE) and human embryonic kidney epithelial (HEK293T) cells up to 100 μM. In contrast, **TBP1** was found to be non-toxic to cancer cells as well as NKE cells up to 100 μM. The non-toxicity towards normal cell lines may be attributed to the low abundance of G4 structures in non-cancerous cell lines. As the proto-oncogenes like *MYC*, *BCL-2* and *c-KIT* are overexpressed in human cancers, the regulatory role of **TBP2** on the proto-oncogenic

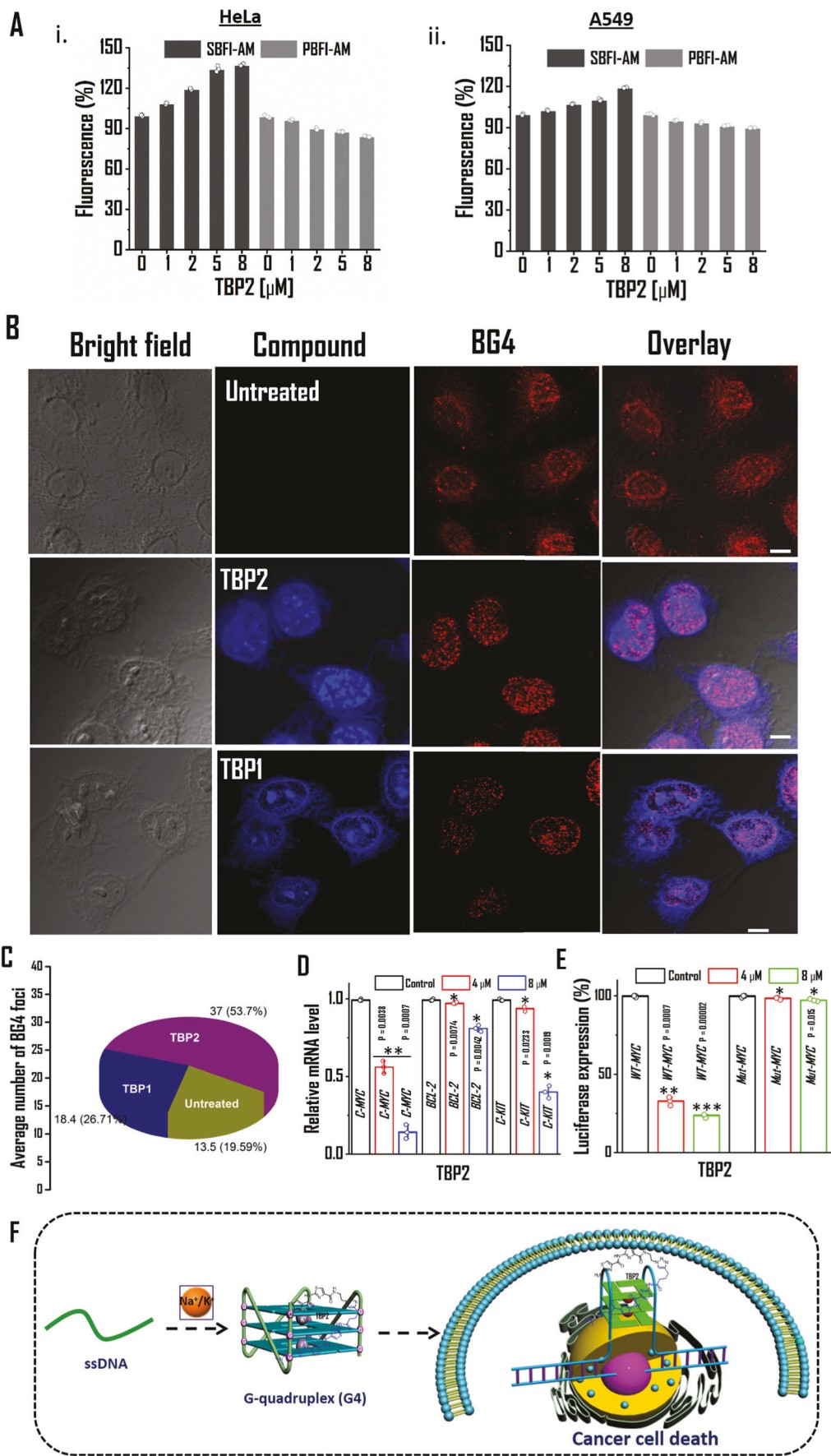

**Fig. 6 | Intracellular activity of TBP2 in cancer cells. A** TBP2 dependent intracellular fluorescence intensity measurements of Sodium-binding benzofuran isophthalate acetoxymethyl ester (SBFI-AM, $Na^+$ probe) and Potassium-binding benzofuran isophthalate acetoxymethyl ester (PBFI-AM, $K^+$ probe) in (i) HeLa and (ii) A549 cells. Error bars represent mean ± SD ($n = 3$). **B** Confocal images of HeLa cell (fixed) stained with **TBP2** (blue) and BG4 (red); scale bars represent 10 μM. **C** Determining the average number of BG4 foci in untreated and ligand treated HeLa cells after 24 h. At least five independent fields were considered to quantify the BG4 foci. **D** qRT-PCR analysis for transcriptional regulation of *c-MYC*, *BCL-2*, *c-KIT* oncogene after treatment with **TBP2** in HeLa cells for 24 h. Quantification was done in terms of fold change by double delta $C_T$ method using 18 s rRNA as a

housekeeping gene. Fold change of ligand treated relative gene expression is normalized with control. Error bars represent mean ± SD ($n = 3$). **P = 0.0038, 0.0007 and *P = 0.0074, 0.0042, 0.0233, 0.0019; as obtained using paired sample statistical analysis (Student's t test), versus untreated or control HeLa cells. **E** Relative luciferase expression of *c-MYC* promoter normalized with the Renilla plasmid pRL-TK after treatment with **TBP2** at two different doses for 48 h. Percentage change of ligand treated relative luciferase expression is normalized with control. Error bars stand for mean ± SD ($n = 3$). **P = 0.0007, ***P = 0.00002; as obtained using paired sample statistical analysis (Student's t test), versus untreated or control HeLa cells.; **F** Schematic illustration of **TBP2**-G4 interaction to mediate cancer cell death. G4; G-quadruplex.

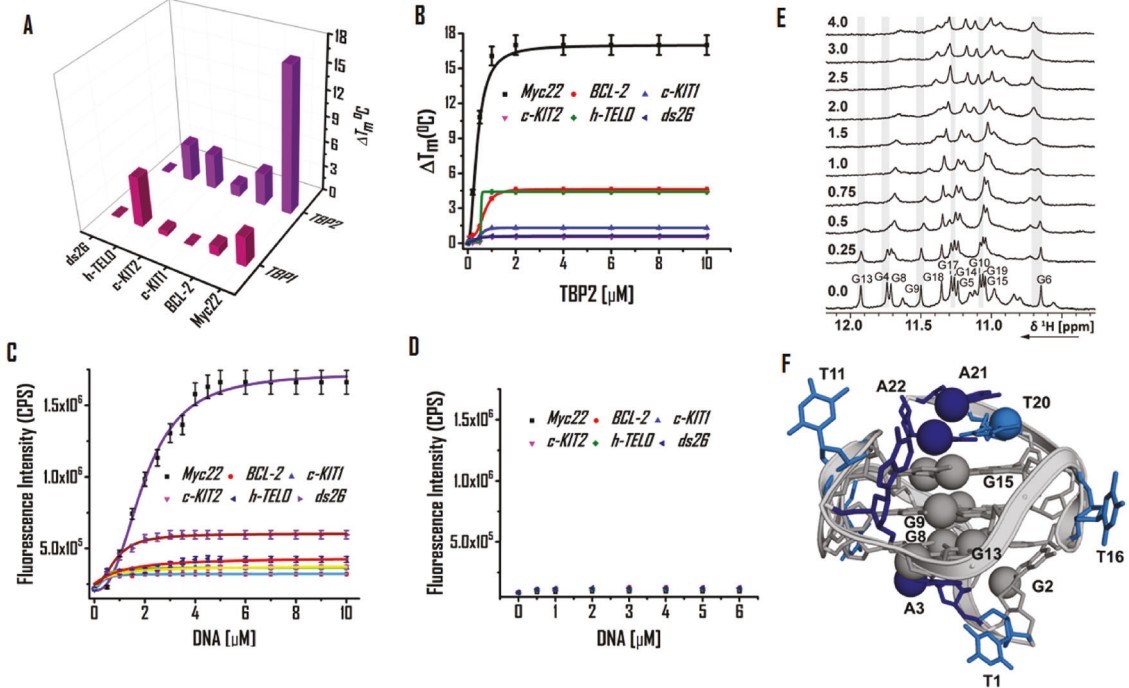

**Fig. 7 | DNA binding properties of TBP2. A** 3D bar diagram of FRET based melting experiments to determine stabilization potential ($\Delta T_M$) of **TBP1** and **TBP2** for G4s and *ds26* DNA. **B** FRET titration of **TBP2** with G4s and *ds26* DNA in 60 mM potassium cacodylate buffer (pH 7.4); Error bars represent mean ± SD ($n = 3$).

Fluorescence titration of (**C**) **TBP2** and (**D**) **TBP1** in 60 mM potassium cacodylate buffer (pH 7.4); Error bars stand for mean ± SD ($n = 3$). 1D 1H NMR spectra of the (**E**) imino region of *c-MYC22* in 25 mM Tris•HCl buffer containing 100 mM KCl at different [**TBP2**:DNA] molar ratios. **F** NMR-structure of *c-MYC22* (pdb-code: 1XAV).

transcription was analysed in HeLa cells (Fig. 6D, see the Supplementary Information, Fig. S12). The qRT-PCR analysis revealed that **TBP2** reduced *c-MYC* mRNA levels by ~44% and ~88% at 4 μM and 8 μM, respectively after 24 h while **TBP1** reduced the relative *c-MYC* mRNA level only up to 42% at 40 μM (see the Supplementary Information, Fig. S12). **TBP2** decreased *c-KIT* expression up to ~6% at 4 μM and ~59% at 8 μM; and *BCL-2* up to ~20% at the highest dose after 24 h as compared to the control HeLa cells (Fig. 6D). Thus, the qRT-PCR data correlated with other biological studies and demonstrated the potential of the peptide mimic **TBP2** to preferentially downregulate the *c-MYC* oncogene expression in cancer cells.

A dual luciferase assay, using wild type and mutant pGL3.0 variants of Del4 *c-MYC* promoter constructs and a renilla plasmid construct (pRL-TK) as the control vector, further confirmed the ability of **TBP2** to alter *c-MYC* oncogene expression in HeLa cells by interaction with G4 harboring promoter region. **TBP2** reduced the luciferase activity of *c-MYC* by ~67% and ~76% at 4 μM and 8 μM, respectively for the wild type promoter after 48 h (Fig. 6E). Intriguingly, **TBP2** did not inhibit the luciferase expression of *c-MYC* mutants. These results demonstrate that **TBP1** was not involved in the regulation of transcription or translation machinery while the synthetic ion transporter, **TBP2** reduced the *c-MYC* oncogene expression through its interaction

with the promoter region of the wild type *c-MYC* luciferase reporter plasmid (Fig. 6F).

## TBP2 preferentially binds to *c-MYC22* G4

As **TBP2** interacts to the promoter region and alters oncogene expression, biophysical assays were conducted to study its binding potential with promoter G-quadruplexes. FRET based melting technique was used to evaluate **TBP1** and **TBP2** triggered stabilization of 5'-FAM and 3'-TAMRA conjugated G4 forming sequences (*c-MYC22*, *c-KIT1*, *c-KIT2*, *BCL-2*, *h-TELO*) and a duplex DNA (*ds26* DNA) by comparing the melting temperature ($T_M$) of control DNAs (without ligand) and ligand treated DNAs. **TBP1** (2 μM or 10 eq. concentration of DNA) did not show significant changes in melting temperatures ($T_M$) for the examined G4s and *ds26* ($\Delta T_M$ values; *c-MYC22* = 3.5 ± 0.2 °C, *BCL-2* = 1 ± 0 °C, *c-KIT1* = 0 °C, *c-KIT2* = 0.5 ± 0 °C, *h-TELO* = 6 ± 0.3 °C, *ds26* DNA = 0 °C; Fig. 7A). Intriguingly, **TBP2** displayed $\Delta T_M$ values of 17 ± 0.9 °C for the *c-MYC22* G4 DNA at 2 μM.

However, mild or no alterations in $\Delta T_M$ values were detected for other G4s and *ds26* DNA ($\Delta T_M$ values; *BCL-2* = 3.8 ± 0.2 °C, *c-KIT1* = 1.3 ± 0.1 °C, *c-KIT2* = 4.1 ± 0.2 °C, *h-TELO* = 4.2 ± 0.2 °C, *ds26* DNA = 0 °C; Fig. 7A, B). These results indicated that **TBP2** selectively stabilized the *c-MYC22* G4 with high stabilization potential ($\Delta T_M$)

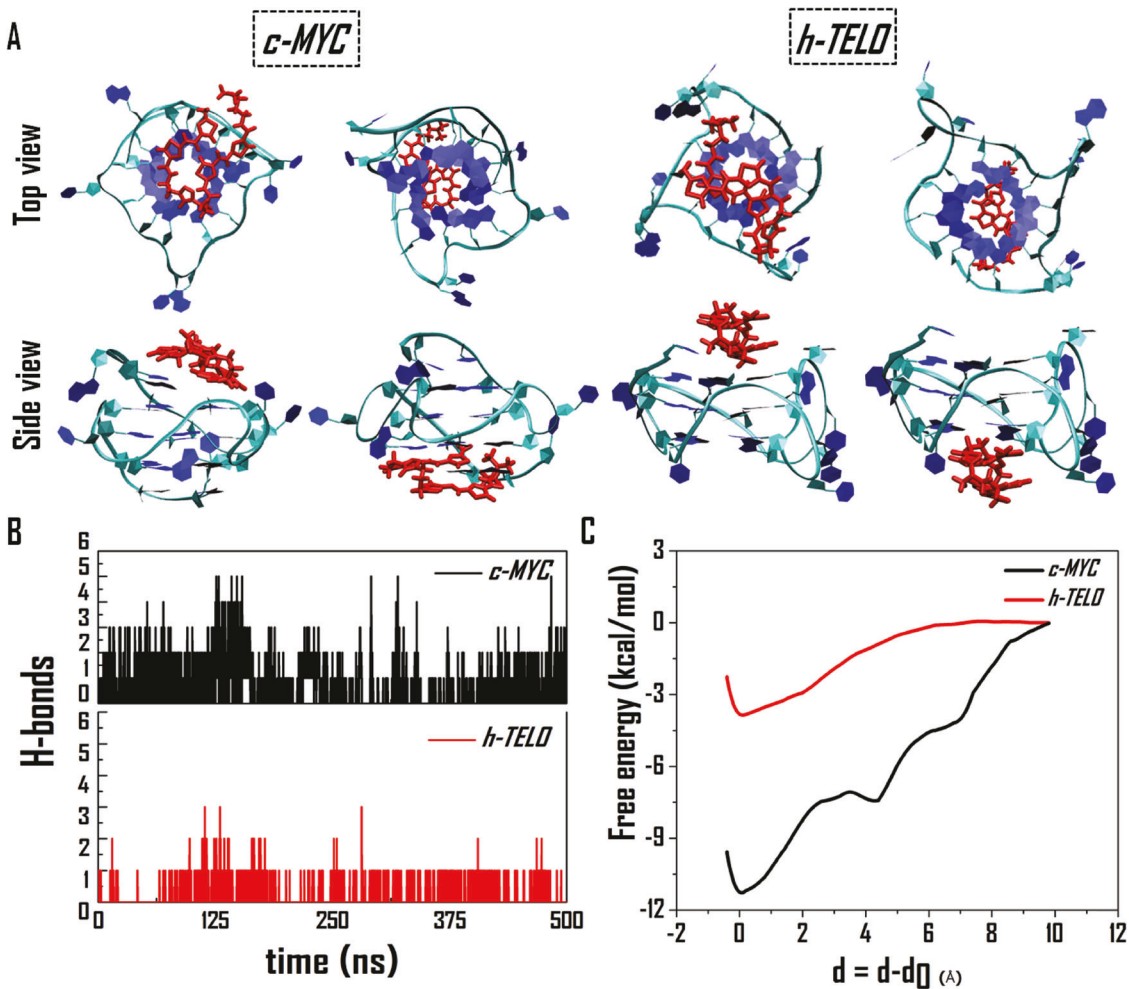

**Fig. 8 | Molecular dynamics simulation of TBP2 with DNA G4s. A** Top and side views of the final structures of **TBP2**, interacting with the top and bottom tetrad of *c-MYC* (1XAV) and *h-TELO* (1KF1) G-quadruplexes, as observed from two MD simulation trajectories. **B** Number of H-bonds between **TBP2** and G-quadruplexes. **C** Free energies of binding between **TBP2** and the two G-quadruplexes. Since the

**TBP2** molecule undergoes adsorption at different distances from the two G-quadruplexes, therefore, for a direct comparison, the reaction coordinate is represented as δ = d-d₀, where, d and d₀ are the distances at any time instant and the initial distance of adsorption, respectively.

values over other G4s and *ds26* DNA. Both **TBP1** and **TBP2** did not show any stabilizing effect for *ds26* (duplex DNA). Fluorometric measurements revealed that **TBP1** and **TBP2** exhibit maxima at 417 nm upon excitation at 325 nm in potassium cacodylate buffer (pH 7.4). A significant enhancement (up to ~ 8 fold) of fluorescence emission intensity of **TBP2** was observed upon gradual addition of pre-annealed *c-MYC22* G4 DNA (Fig. 7C, see the Supplementary Information, Fig. S17B), while no noticeable changes in its intensity were observed with other DNA quadruplexes like *c-KIT1*, *c-KIT2*, *BCL-2*, *h-TELO* and *ds26* DNA (Fig. 7C, see the Supplementary Information, Fig. S17). **TBP2** exhibited a dissociation constant ($K_d$) of 1.6 µM for the *c-MYC22* G4 (Fig. 7C) and higher $K_d$ values for *c-KIT1* (9.5 µM), *h-TELO* (8.2 µM) and *ds26* DNA (8.2 µM), illustrating preferential binding of **TBP2** to *c-MYC* G4 over other experimented G4s and *ds26* DNA. Peptidomimetic **TBP1** did not show affinity towards G4s or *ds26* DNA as revealed from fluorescence titrations (Fig. 7D, see the Supplementary Information, Fig. S17).

The interaction of **TBP2** with *c-MYC22* was investigated by recording 1D 1H NMR spectra at different [**TBP2**]:[DNA] ratios (Fig. 7E, F, see the Supplementary Information, Fig. S18). The titrations were performed in 25 mM Tris•HCl buffer containing 100 mM KCl. The *c-MYC22* DNA formed a definite G-quadruplex structure that was described by Yang et al. (PDB: 1XAV). The DMSO effect on the *c-MYC22*

G-quadruplex was previously discussed[62] and the assignment for the target DNA was followed from Yang et al.[63]. Clear changes in the 1D 1H spectrum upon addition of the ligand to the G-quadruplex formed by the sequence *c-MYC22* were detected. The addition of 0.25 equivalents **TBP2** led to general line broadening of the imino signals (Fig. 7E). However, further addition revealed stronger effects for the imino signals of G4, G9, G10, G13 and G17. For the imino signal of G6, a second signal for the bound state seemed to appear that was present in a 1:1 ratio at a [ligand:DNA] molar ratio of 1. In the aromatic region, the signals of A3, G2 or G4, G5, G13, T20, A21H2, A22, A22H2 and some signals of the overlapping region were mostly affected by ligand-binding (see the Supplementary Information, Fig. S18). The imino and aromatic signals showing the largest changes upon addition of **TBP2** are marked in red and with spheres on the NMR-structure of *c-MYC22* (pdb-code 1XAV). The data demonstrates that the strongest effect was observed for nucleotides on the lower and upper tetrad as well as both capping structures and G5 and G9 from the middle tetrad (Fig. 7E, F, see the Supplementary Information, Fig. S18). Thus, the interaction of **TBP2** took place on the bottom and top of the G-quadruplex and there was probably a direct interaction with G6 as well.

We further employed classical MD simulations to gain insights and compare how **TBP2** interacts with two topologically distinct G4s *c-MYC22* (PDB: 1XAV) and *h-TELO* (PDB: 1KF1). For each of these G4s, two

sets of MD simulations were conducted, placing the **TBP2** molecule -12 Å away from both the upper and lower tetrads of the quadruplex. Figure 8A displays the top and side views of the final composite structures for *c-MYC* and *h-TELO* G4s in both sets of simulations. The simulations results revealed that the **TBP2** molecule binds to the upper and bottom tetrad motif of both the G4s; aligning with NMR findings. Further analysis showed that in the case of *c-MYC*; A2, G4, G5, G13, G17, T20, A21, and A22 residues interacted with **TBP2**, while for *h-TELO*; G2, G3, G4, T5, T11, G15, T17, G21, and G22 were involved in the interaction with **TBP2**.

The assessment of hydrogen bonds revealed that *c-MYC* formed 2-3 hydrogen bonds with **TBP2**, whereas *h-TELO* only formed a single hydrogen bond (Fig. 8B). However, the connections between **TBP2** and the G4s mainly originated from van der Waals and long-range electrostatic interactions. The free binding energies of **TBP2** were −11.5 kcal/mol for *c-MYC* and −3.9 kcal/mol for *h-TELO* G4s, suggesting a more favorable binding affinity with *c-MYC*, consistent with our biophysical findings. The simulation images indicate that **TBP2** gets fitted onto the top or bottom of *c-MYC*, resulting in a higher binding energy, while the fitting cavity of *h-TELO* could not adequately accommodate the **TBP2** molecule, leading to a comparatively weaker interaction (Fig. 8A).

In summary, we have designed a peptide mimic that forms vesicular and nanofibrillar non-covalent framework in response to specific microenvironments. The supramolecular nanostructures of **TBP2** can span the lipid membrane, displaying distinct channel behavior in the presence of Na⁺ or K⁺, thereby regulating intracellular ion concentration. **TBP2**, possibly in its monomeric form, enters cell membranes and binds to non-canonical DNA structures in cell nuclei, and reduces cellular transcription.

Moreover, **TBP2** rapidly localizes in model lipid bilayers and the plasma membrane to facilitate ion transport, while it accumulates in the nuclei of HeLa cells after prolonged duration (e.g., 24 h) in a time-dependent manner, showing enhanced fluorescence in cellular system. This versatility makes **TBP2**, a promising molecular scaffold for studying membrane related and intracellular processes. The formation of channels by **TBP2,** increase intracellular concentration of monovalent cations like Na⁺, the essential component of DNA G4 structures. The time-dependent membrane or nuclear localization, subsequent increased intracellular cation concentration and direct interaction with G4 structures promote the formation and stabilization of G4s, leading to cancer cell death. This class of synthetic peptide nanostructures provides structural and functional insights into ion channels and represents a paradigm for developing artificial transporters with therapeutic potential.

## Methods
### TEM imaging
TEM experiments were carried out in bright-field mode on a JEOL 1200 EX electron microscope, operated at an acceleration voltage of 120 keV. **TBP1** or **TBP2** were diluted in HEPES NaCl or HEPES KCl buffer (pH 7.4) at the con. of 20 μM. The sample was prepared by placing a drop (5 μL) of aqueous dispersions of **TBP1** and **TBP2** on a carbon-coated copper grid and air-dried at room temperature overnight.

### Ion transport study by fluorescence spectroscopy
Large unilamellar vesicles (LUVs) were formed using a 200 μL 9:1 mixture of 10 mM EYPC (L-α-Phosphatidylcholine egg yolk) and cholesterol in chloroform. After solvent removal and vacuum drying, the resulting thin film was hydrated with 500 μL of buffer (10 mM HEPES, 100 mM NaCl or KCl, pH 6.4) containing 100 μM HPTS (8-hydroxypyrene-1,3,6- trisulfonic acid trisodium salt). Next, the suspension was subjected to six freeze–thaw cycles (liquid nitrogen/water at room temperature) during hydration. The resulting white suspension was then extruded 19 times through a 100 nm polycarbonate membrane to obtain large unilamellar vesicles (LUVs) with an average diameter of ~ 60 nm (as measured by DLS method). The LUVs suspension was

separated from extravesicular HPTS dye by using size-exclusion chromatography (Econo-Pac 10DG column, Bio-rad; mobile phase: 10 mM HEPES, 100 mM NaCl or KCl, pH 6.4) and diluted with mobile phase for the desired working concentration.

**TBP1** or **TBP2** (0–20 μM) was added to a EYPC· LUVMs⊃ HPTS suspension ([EYPC] = 10 mM, [HPTS] = 100 μM) in 10 mM HEPES buffer containing 100 mM MCl [M = Na⁺, K⁺, Cs⁺, Rb⁺, Li⁺] (490 μL, pH 6.4) followed by subsequent addition of an aqueous solution of NaOH (0.5 M, 5 μL, ΔpH = 1) in a clean and dry fluorescence cuvette. Fluorescence intensity of HPTS at 510 nm upon excitation with 460 nm·light was monitored as a function of time until the addition of 1.0 wt% Triton X-100 (40 μL) at 450 s. Relative fluorescence intensity of HPTS was evaluated by the equation of

$$I = \frac{I_t - I_0}{I_{Lysed} - I_0}$$

Where $I_O$, and $I_t$ represent the initial, final and $I_{Lysed}$ represent the fluorescence intensities before addition of NaOH, after addition of NaOH and 1 wt% Triton X as lysis buffer, respectively.

The Hill equation was used to determine the $EC_{50}$ values (concentration of molecules to achieve 50% ion transport activity) of the peptidomimetics:

$$I = \frac{1}{1 + \left(\frac{EC_{50}}{[Channel]n}\right)}$$

I = relative fluorescence intensity, [Channel] = concentration of **TBP1** or **TBP2**, n = Hill co-efficient

### Formation of giant unilamellar vesicles (GUVs)
GUVs were prepared by electroformation technique (Vesi Prep Pro, Nanion, Germany). 10 μL of a 10 mM solution of EYPC and cholesterol (9:1) in chloroform was spread evenly on the indium tin oxide (ITO) coated glass slides within the "O" ring area. The solvent was evaporated at room temperature and the slides were dried under vacuum. Then, ITO slides were assembled in the Vesi Prep Pro and filled with 275 μL of sorbitol solution (1 M). A sinusoidal AC field of 3 V and 5 Hz was applied for 2 h at 25 °C temperature. The prepared GUV solution was collected and subjected to patch-clamp experiments.

### Confocal imaging in GUVs and cells
GUVs were suspended in 10 mM HEPES, 100 mM KCl or 100 mM NaCl (pH 7.4) buffer and incubated with **TBP1** or **TBP2** (20 μM) and nile red (Sigma, USA) for 5 minutes. After incubation, the mixture was placed on the fluodish and observed under Confocal Microscope (Zeiss, Germany). For control slides, GUVs were incubated with either **TBP2** or Nile red. At least 4 fields per slide and three independent sets were examined. HeLa cells were treated with **TBP1** or **TBP2** (8 μM) following nile red (10 μM) incubation 10 minutes prior to imaging. The images obtained were processed using ImageJ software.

### Conductance measurements by automated patch-clamp technique
Conductance measurements were carried out using the Port-a-Patch setup (Nanion, Munich, Germany). First, a borosilicate glass chip (NPC chip, Nanion, Germany) with 3−5 mΩ was loaded with symmetrical working buffer containing 10 mM HEPES 1 M MCl [M = Na⁺, K⁺, Cs⁺, Li⁺, Rb⁺] buffers (pH 7.0) in both *cis* and *trans* compartments and Ag/AgCl electrodes were placed on both sides of the NPC chip. Next, bilayer membrane with >1 Giga Ohm resistance was constructed across the micrometer-sized aperture in the NPC chip by adding GUV suspension and applying a mild negative pressure (−10 mbar). **TBP2** (20 μM) was added to the *cis*-side of the chamber. Current traces were recorded using an HEKA EPC 10 patch clamp amplifier with a built-in 1 kHz 4 pole

Bessel low-pass filter and a Digidata 1322 A digitizer. I-V curve was generated using a voltage ramp from -80 mV to +80 mV. Data analysis was performed using Clampfit 10.2 software.

**Measurement of intracellular $Na^+$ and $K^+$ concentration**

HeLa or A549 cells were cultured in 96 well plates and incubated with 10 μM fluorescent $Na^+$ probe [sodium-binding benzofuran isophthalate acetoxymethyl ester (SBFI-AM)] or $K^+$ probe [potassium-binding benzofuran isophthalate acetoxymethyl ester (PBFI-AM)] in culture media for 1.5 h at 37 °C. Cells were washed with 1x PBS to remove the free extracellular SBFI-AM or PBFI-AM following treatment with **TBP2** at different concentrations (0, 1, 2, 5, 8, 20 μM) with 2 h incubation period at 37 °C. The fluorescence of SBFI-AM ($\lambda_{ex} = 350$ nm, $\lambda_{em} = 527$ nm) or PBFI-AM ($\lambda_{ex} = 350$ nm, $\lambda_{em} = 527$ nm) was measured in a microplate reader (Molecular Devices, USA).

**NMR titration**

The *c-MYC22* DNA was purchased from Eurofins MWG Operon in HPSF grade followed by purification with HPLC. The titration was carried out with 100 μM DNA in 25 mM Tris•HCl buffer (pH 7.4), which contains 100 mM KCl in 5% d6-DMSO/95% $H_2O$. Low volume of **TBP2** stock solution in 100% d6-DMSO was added directly into the NMR tube (7.5% d6-DMSO at the end of the titration). 2,2-dimethyl-2-silapentane-5-sulfonate (DSS) was kept as reference. For water suppression, gradient-assisted excitation sculpting or jump-return-Echo was used.

**Reporting summary**

Further information on research design is available in the Nature Portfolio Reporting Summary linked to this article.

## Data availability

The data that support the findings of this study are available in the manuscript and Supplementary Information file. The reported PDB codes and crystal structures of *c-MYC* (1XAV) and *h-TELO* (1KF1) are given in the manuscript. The atomic coordinates of the optimized computational models are deposited in the Zenodo OpenAIRE database under doi: 10.5281/zenodo.11118221. All other information is available from the corresponding authors upon request. Source data available with this manuscript. Source data are provided with this paper.

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

## Acknowledgements

R.P. thanks *DST-India* for *INSPIRE* fellowship [IF160996]. D.D. and B.L. thank *CSIR-India* for a research fellowship. The authors thank Mrs. Debapriya Ghatak, IACS for Confocal Microscopy. We thank Sri Gopal Manna, IACS for the 3D artwork. This work was supported by *Wellcome Trust-DBT India Alliance* [Grant Number, IA/S/18/2/503986] fellowship grant and *DST-SERB CRG* research grant [CRG/2021/004525]. We also thank Technical Research Committee (TRC) grant, IACS for the additional funding.

## Author contributions

The manuscript was written through contributions of all authors. J.D. conceived the study; J.D. and R.P. designed the experiments; J.D. and R.P. wrote the paper; D.D. and B.L. synthesized the molecules; R.P. conducted the biophysical and biological experiments; T.M. and A.D. performed the quantum chemistry calculations and molecular dynamics simulations; D.M. and H.S. contributed to the NMR experiments and data analysis. All authors have given approval to the final version of the manuscript.

## Competing interests

The authors declare no competing interests.
