## [Peer Review File · Nature Communications]

Expanding the landscape of ion transporters: A non-B DNA binding peptidomimetic channel alters cellular functionsReviewer #1 (Remarks to the Author):

In this manuscript, Dash et al. reported the thiazole-based peptide-like molecule TBP2 which allows transportation of Na⁺ and K⁺ through the lipid bilayer membranes, and also permeates into nuclei to interact with DNA G-quadruplex to induce cell death. However, although the authors claim the novelty of TBP2 having dual functions, which may indeed be of some interest, namely ion transport and regulation of gene function, it is difficult to see the practical advantage of this dual functionality in biological applications. In addition, the analysis of the characteristics of TBP2 is also not sufficient to support the authors' conclusions. Given these considerations, this manuscript would not be suitable for publication in Nature Communications.

Specific Comments

- 1) In the DLS data in Figure 1B, the distribution of the hydrodynamic diameter should be shown with a log-based x-axis including a larger size range. It would also be preferable to use histograms for presenting the size distributions.
- 2) In fact, it is curious that large particles, corresponding to the nanofibers, are not detected by the DLS measurement. The DLS profile possibly shows a multimodal distribution of different-size particles. Did the authors separate the vesicles from the nanofibers? If so, how did they do it?
- 3) Is it possible to visualize the co-localization of TBP2 in the model membranes by fluorescence microscopy using GUV?
- 4) Vesicles are usually nano-sized, and it is not necessary to use the term "nanovesicle".
- 5) What about the transport of other cations like Li⁺, Cs⁺, or Rb⁺. Also, do the divalent cations could also be transported by TBP2?
- 6) The results of the molecular simulation suggested the formation of nanofibers. Then, how could the simulation explain the formation of vesicles? How do the TBP2 molecules align to form the vesicle?
- 7) The calculated binding energies between TBP2 and K⁺ or Na⁺ are significantly high. Are these results consistent with the experimentally evaluated binding constants of TBP2 with K⁺ and Na⁺, respectively?
- 8) Based on the results of the ion transport assays presented, it is difficult to conclude that the TBP2 molecules form the ion channels in the membrane. It is also possible that TBP2 molecules acts as mobile carriers. Further experimental evidence is needed to prove that they form the ion channels.
- 9) Why did the ion conductance profiles in Figures 3A and 3C or Figures 3B and 3D differ significantly from each other, while the I-V plots in Figure 3G show the linear profiles.? From these results, it is difficult to conclude that the ion channels are formed in the membrane.
- 10) Does Figure 4A show the initial structure of the ion channel formed by TBP2 in the simulation? As a result of MD simulation, the alignment of TBP2 molecules seems to be too regular. The authors should provide a movie of the MD simulation.
- 11) What about the effect of TBP2-induced ion permeation on the cell viability?
- 12) Figure 5A does not show the localization of TBP2 molecules to the nuclei, but indicates that they are present in both the nuclei and cytoplasm.
- 13) The difference in the dissociation constants between c-MYC22 G4, c-KITI, h-TELO, ds26 DNA and TBA2 is less than 10-fold and not very large, whereas the difference in the fluorescence intensity change during titration between c-MYC22 G4 and others is significant. What is the reason for this?
- 14) It is necessary to explain why TBA2 shows a particularly strong interaction with c-MYC22 G4 than others.

Reviewer #2 (Remarks to the Author):

In this manuscript Dash and coworkers describe the synthesis and the biophysical characterisation of a peptide-mimic (TBP2) and its cellular properties as a G4-binder. The manuscript is effectively split in 2 parts. In the first part, the authors described the biophysical properties of TBP2 in terms of aggregation and as a peptide-like ion channel in artificial membranes, which could lead to K⁺

transport in artificial cells. This part of the manuscript is very well described and characterised, with a series of biophysical techniques used to characterise the ion channel properties of TPB2. In the second part of the manuscript the authors apply TPB2 in cells and study their effect on G4-stabilisation. There is more confusion here and it is hard to understand if the authors are implying that TPB2 binds as a single entity to nuclear G4s or if the molecule induces K⁺ increases in cells, thus causing formation of G4s. The authors then go back to biophysics to show that TPB2 has a preferential binding to MYC G4 and speculate that this is the reason why they observe gene-expression downregulation using a luciferase assay.

Overall, my main issue is that these 2 parts of the manuscript are very disjoint and it is hard to understand how they are linked, they sound more like two separate manuscripts. Furthermore, the second part is very speculative and not enough evidence is provided to justify the mechanisms suggested in Figure 5. Based on this, I cannot be supportive of this manuscript in its current form for publication in Nat Commun.

Below are my main concerns:

- 1) There is no link between the biophysical properties of the TPB2 as an ion channel and its cellular activity. The authors imply in Figure 5 that there is an increase in K⁺ levels but they provide no evidence to support this. How the biophysical characterisation of TPB2 as an ion channel is relevant for its G4-binding property as a monomer is not clear at all. It reads to me like two separate studies that have been merged somehow.
- 2) The confocal microscopy clearly shows wide TPB2 distribution in the nuclei of human cancer cells (Figure 5), so even if biophysically there is some selectivity for binding MYC G4, this is clearly not relevant under the conditions used for cellular experiments, as revealed by BG4 co-localisation. So how the MYC selectivity data is relevant is hard to appreciate.
- 3) On a similar note to point 2 the authors use a luciferase assay with MYC promoter to assess the ability of TPB2 to bind MYC G4, which is fine, but clearly cannot be used as an indicator of selective binding for this particular G4 in cells. RT-qPCR data should be collected looking also at other promoters containing G4s (i.e. KIT, SRC etc) and show preferential down regulation of MYC, if the authors want to claim selectivity.
- 4) The authors show BG4 foci increase upon treatment with TPB2, so again their evidence suggests broad G4-stabilisation. If the authors want to speculate this is due to the localisation of the molecule in the plasma membrane of the HeLa cells then they should be able to quantify K⁺ concentrations increasing after treatment with the molecule. Otherwise it is hard to distinguish the effect of stabilisation of the monomer vs the ion channel effect (if any).

Overall this is a potentially interesting manuscript with some interesting preliminary evidence. However, currently the data reported are scattered and disjoint not really allowing the authors to support their conclusions. I would recommend to either split this manuscript into 2 smaller articles or to accumulate more evidence to fully support the authors' speculation on the cellular mechanism of action.

Point-by-point response to the Reviewers' comments:

Reviewer #1 (Remarks to the Author):

In this manuscript, Dash et al. reported the thiazole-based peptide-like molecule TBP2 which allows transportation of Na⁺ and K⁺ through the lipid bilayer membranes, and also permeates into nuclei to interact with DNA G-quadruplex to induce cell death. However, although the authors claim the novelty of TBP2 having dual functions, which may indeed be of some interest, namely ion transport and regulation of gene function; it is difficult to see the practical advantage of this dual functionality in biological applications. In addition, the analysis of the characteristics of **TBP2** is also not sufficient to support the authors' conclusions.

We appreciate the reviewer for recognizing the novelty and significance of our study regarding the dual functionality of the peptide mimic **TBP2**. We would like to emphasize that the innovative part of our work lies in unifying synthetic peptide-based ion channels with distinctive intracellular properties for applications in nanobiotechnology and molecular and biomedical engineering. This approach would allow us to mimic and regulate complex biological functions, thereby presenting potential therapeutic tools for challenging diseases like cancer. We acknowledge the reviewers' suggestions regarding the need of important experiments and data analysis to substantiate our findings.

The dimeric thiazole peptide **TBP2** mimics DNA binding proteins and consists of peptide (CO=NH) linkages, hydrophobic aromatic parts and a hydrophilic -NMe₂ group. These units are crucial for maintaining structural stability, facilitating hydrophobic interactions with lipid membranes and also contributing to DNA binding.

It is noteworthy to state that we conducted the ion channel experiments (e.g., HPTS assay, CF release assay, patch clamp studies, membrane colocalization) on model lipid bilayers instantly following the incorporation of **TBP1** and **TBP2** ligands. We have established the molecular basis of ion channel formation through theoretical investigations, including quantum chemistry calculations and molecular dynamics simulations. Conversely, the effect of the **TBP1** and **TBP2** was assessed after a minimum of 24 h of treatment in both cancer and normal kidney epithelial cells. The cell viability study suggests that **TBP2** can induce cancer cell death [IC₅₀ values of ~8.2, ~10 and ~16 μM in cervical carcinoma (HeLa), leukemia (K562) and lung carcinoma (A549) cells, respectively], while normal kidney epithelial (NKE) cells remain unaffected up to 100 μM. However, **TBP1** did not induce cell death in either cancer or normal cells up to 100 μM as determined via the XTT based cell viability assay. We conducted fluorescence experiments to examine the changes in intracellular Na⁺ or K⁺ concentrations induced by **TBP2** using specific fluorescent dyes of e.g., sodium-binding benzofuran isophthalate acetoxymethyl ester (SBFI-AM, Na⁺ probe) and potassium-binding benzofuran isophthalate acetoxymethyl ester (PBFI-AM, K⁺ probe). HeLa or A549 cells were incubated with SBFI-AM or PBFI-AM (10 μM) for 2 h under standard culture conditions, followed by treatment with varying concentrations of **TBP2** (ranging from 0 to 8 μM) for 1.5 h.

The intracellular fluorescence measurements indicated that **TBP2** (8 μM) significantly increased Na⁺ concentration by ~ 37% and marginally decreased the cytosolic K⁺ concentration ~ 15% in HeLa cells (Figure 5E). In A549 cells, **TBP2** led to a ~19% increase in Na⁺ concentration and a ~10% decrease in K⁺ concentration compared to the control. These results could be rationalized by different Na⁺ or K⁺ distributions across the cell membranes as the intracellular Na⁺ concentrations (~12 mM) are much lower than the extracellular Na⁺ (~145 mM), whereas the extracellular potassium concentrations (~4 mM) are much lower than its intracellular counterpart (~150 mM) under physiological conditions. Therefore, the data indicate that the **TBP2**-induced intracellular increase in Na⁺ could have disrupted the ionic balance, potentially synergizing the formation and broad stabilization of G-quadruplex structures. This effect might have occurred following the accumulation of **TBP2** over an extended duration within cancer cells. Furthermore, we have also established that **TBP2** has the ability to downregulate oncogene expression via a G-quadruplex (G4)-dependent mechanism, which could potentially contribute to the initiation of cancer cell death. The G4 based mechanistic study was carried out using different biophysical (FRET based melting assay, fluorescence spectroscopy, NMR titration) and biological experiments (qRT-PCR, confocal microscopy for immunofluorescence assay with G4-specific antibody BG4, G4 specific promoter activity assay or dual luciferase reporter assay).

We have explored the structure and function of the supramolecular arrangement of **TBP2** through theoretical investigations. Quantum chemistry calculations followed by classical molecular dynamics simulations were performed to optimize and extend the structure of a single **TBP2** molecule to a supramolecular arrangement of six molecules (Figure 1 & S7, Supplementary Information). These molecules were shown to form a vertically stacked arrangement, forming a channel which possessed enough pore size to allow the passage of sodium and potassium ions. Further, the supramolecular system was embedded into a lipid bilayer and demonstrated to perform spontaneous ion transport (Figure 4) in our molecular dynamics simulations. Free energy calculations were performed to calculate the energy barrier for the passage of cations (Figure 4) through the pore, which clearly supported the experimental observations of ion passage and higher affinity towards cations.

To the best of our knowledge, we report a first example of a G4 binding peptidomimetic that forms the artificial ion channel in the lipid bilayer which could play significant contribution in the field of interdisciplinary sciences and inspire the further development of synthetic ion transporters with biological implications.

Specific Comments

1) In the DLS data in Figure 1B, the distribution of the hydrodynamic diameter should be shown with a log-based x-axis including a larger size range. It would also be preferable to use histograms for presenting the size distributions.

- In the revised manuscript, the size distribution of vesicles is now presented as histogram with log-based x-axis (Figure 1B).

Figure 1B. The average size distribution (\pm SD) and abundance of **TBP2** formed vesicles as measured by DLS method.

2) In fact, it is curious that large particles, corresponding to the nanofibers, are not detected by the DLS measurement. The DLS profile possibly shows a multimodal distribution of different-size particles. Did the authors separate the vesicles from the nanofibers? If so, how did they do it?

-The authors thank the reviewer for concerns related to the multimodal distribution of different-sized particles by dynamic light scattering (DLS) method. We agree with the reviewer that nanofibers could interfere with the vesicles size determination by DLS. The high resolution TEM images revealed **TBP2**-formed nanofibres are more abundant than vesicles in NaCl buffer solution; the vesicles were separated by centrifugation method at lower centrifugal force (1000g) and the supernatant (vesicular solution) was subjected to DLS experiment.

Figure 1. (ii)(A) TEM image of **TBP2** showing distribution of vesicles (indicated with purple arrow). B) The average size distribution (\pm SD) and relative abundance of **TBP2** formed vesicles as measured by DLS method.

3) Is it possible to visualize the co-localization of **TBP2** in the model membranes by fluorescence microscopy using **GUV**?

- Yes, it is possible to visualize the co-localization of **TBP2** in the model membranes using fluorescence microscopy with **GUV** (Giant Unilamellar Vesicles). The fluorescence microscopy-based images showing the membrane co-localization of **TBPs** are presented in the main manuscript (Figure 2J).

Figure 2J. Membrane colocalization of **TBP2** and **TBP1** with Nile Red in GUVs. Scale bars represent 5 μm (top row) and 2 μm (bottom row).

Confocal microscopy based imaging experiments were conducted using GUVs stained with Nile Red to assess the membrane embedding feature of **TBP1** and **TBP2**. These GUVs were generated from EYPC lipid using electroformation technique. Microscopic imaging results indicated that **TBP2** embedded within GUV membrane and co-localized with the membrane staining dye Nile Red (Figure 2J). Moreover, **TBP1** disrupted the membrane structure while **TBP2** preserved the membrane integrity of the vesicles (Figure 2J), consistent with the CF release assay.

4) Vesicles are usually nano-sized, and it is not necessary to use the term “nanovesicle”.

‘Nanovesicles’ has been changed to ‘vesicles’.

5) What about the transport of other cations like Li^+ , Cs^+ , or Rb^+ . Also, do the divalent cations could also be transported by **TBP2**?

The authors thank the reviewer for raising concerns regarding **TBP2**-mediated transport of other alkali cations like Li^+ , Cs^+ , or Rb^+ . In response to this suggestion, we conducted further HPTS assays to investigate **TBP2**'s role in transporting Li^+ , Cs^+ , or Rb^+ using HEPES- LiCl , HEPES- CsCl or HEPES- RbCl buffers (Figure S1.2). The HPTS data demonstrated that **TBP2** exhibited EC_{50} values of $54 \pm 2.7 \mu\text{M}$, $26 \pm 1.3 \mu\text{M}$ and $14 \pm 0.7 \mu\text{M}$ for Li^+ , Cs^+ , or Rb^+ , respectively; indicating its greater transport efficiency for K^+ ($\text{EC}_{50} = 5.8 \pm 0.3 \mu\text{M}$) and Na^+ ($\text{EC}_{50} = 7 \pm 0.4 \mu\text{M}$). Based on HPTS data, the transport efficiency of **TBP2** for various alkali cations can be inferred as $\text{K}^+ > \text{Na}^+ > \text{Rb}^+ > \text{Cs}^+ > \text{Li}^+$. However, the EC_{50} value for transporting divalent cation e.g., Ca^{2+} , was determined to be $135 \pm 6.8 \mu\text{M}$. This indicates a low ionophoric activity of **TBP2** for divalent ions.

Figure S1.2. **TBP2** gated ion transport via model lipid bilayer. (A-C, E) HPTS assay for measuring the ion transport activity of **TBP2** in the presence of Cs^+ , Li^+ , Rb^+ and Ca^{2+} ions. Buffer composition: Internal –10 mM HEPES 100 mM NaCl (pH 6.4), External –10 mM HEPES 100 mM CsCl , LiCl , RbCl or CaCl_2 (pH 7.4). (D, F) EC_{50} value determination of **TBP2** in the presence of external buffer – 10 mM HEPES 100 mM CsCl , LiCl , RbCl or or CaCl_2 (pH 7.4), internal buffer – 10 mM HEPES 100 mM NaCl (pH 6.4).

We further conducted electrophysiology experiments to evaluate the channel behavior of **TBP2** using GUVs in the presence of KCl , NaCl , LiCl , CsCl or RbCl buffers. The *cis* and *trans* compartments, each containing a 1 M MCl [$\text{M} = \text{K}^+$, Na^+ , Rb^+ , Cs^+ or Li^+] solution, were separated by a planar lipid bilayer membrane composed of EYPC lipid. Currents were recorded over time under various positive and negative

applied potentials (Figure 3). The average conductance for transporting K^+ , Na^+ , Rb^+ , Cs^+ or Li^+ was measured to be 0.68 ± 0.03 , 0.56 ± 0.03 , 0.29 ± 0.01 , 0.15 ± 0 , and 0.081 ± 0 nS, respectively; suggesting **TBP2** transported K^+ or Na^+ with high conductance compared to Rb^+ , Cs^+ or Li^+ under similar applied voltages (Figure 3, Supplementary Information, Figure S4).

Figure 3. Bar diagram for frequency vs current of **TBP2** in the presence of 1 M NaCl at (E) -80 mV, (H) +80 mV or 1 M KCl at (F) -80 mV, (I) +80 mV. (G) I-V plot of **TBP2** in the presence of 1 M MCl [$M = Na^+$, K^+ , Cs^+ , Li^+ , Rb^+] buffers. (J) Drop line plot of conductance of **TBP2** for different metal ions.

6) The results of the molecular simulation suggested the formation of nanofibers. Then, how could the simulation explain the formation of vesicles? How do the **TBP2** molecules align to form the vesicle?

The authors appreciate the reviewers comment. As outlined in the manuscript, the high resolution transmission electron microscopic imaging revealed the presence of both nanofibers and vesicles, while the molecular dynamics simulation study provided insights into the supramolecular arrangement and molecular basis of **TBP2** gated ion channel formation through the lipid membrane.

Both hydrophobic and hydrophilic parts of the compounds play important role in the process of forming various vesicular structures. The formation of these vesicular structures likely arose from the combination of supramolecular self-assembly and the propensity for robust aggregation of the aromatic surfaces of **TBPs**. Peptidomimetic **TBP1**, with the Boc protecting group was unable to generate ordered structures like **TBP2** in the presence of Na^+ or K^+ (see the Figure S6, SI). While TEM images of **TBP1** displayed a limited number of vesicular structures, they were not as distinguishable as those observed with **TBP2**. Consequently, the $-NH_2$ group present in **TBP2** played a crucial role in forming ordered non-covalent arrangements.

7) The calculated binding energies between **TBP2** and K^+ or Na^+ are significantly high. Are these results consistent with the experimentally evaluated binding constants of **TBP2** with K^+ and Na^+ , respectively?

The binding energies were calculated using two different functionals M06-2x and B3LYP in vacuum phase, and both of the functionals produced consistent binding energies. DFT tends to slightly overestimate the binding energies for systems containing ions in a vacuum, as the charge effect remains unscreened by solvent influences. Therefore, while the binding free energies may seem relatively high, the trend remains consistent both in a solvent phase and in experimental conditions. To further validate the results obtained from DFT, in the revised manuscript we conducted MD simulations to calculate the binding free energies of ions with individual **TBP2** molecules in a vacuum, which allows us not only to compare the magnitudes

with the DFT-obtained results, but also to validate the quality of the MD parameters used in this study. It is noteworthy to state that the binding free energies obtained from MD simulations contain both enthalpic and entropic effects. In contrast, those obtained from DFT calculations accounts only the enthalpic effect, as the temperature and dynamic effects are absent in DFT calculations. The calculated binding energies of Na⁺, and K⁺ were -59.6 and -47.5 kcal/mol respectively, which clearly indicates a significant similarity in the enthalpic contribution compared to the DFT-calculated values, given that the binding process reduces entropy, leading to a negative -TΔS term. In a solution, the binding energies are expected to further decrease due to the screening effect of the solvent molecules, which reduces the binding enthalpy and also contributes towards the entropic effect.

8) Based on the results of the ion transport assays presented, it is difficult to conclude that the TBP2 molecules form the ion channels in the membrane. It is also possible that TBP2 molecules act as mobile carriers. Further experimental evidence is needed to prove that they form the ion channels.

-The authors acknowledge the reviewer's concern regarding the conclusiveness of **TBP2** molecules forming ion channels in the membrane. The ion channel experiments are primarily based on the fluorescence based assays (e.g., HPTS assay, CF release assay, Cl⁻ specific lucigenin assay, safranin O based membrane polarization assay) and electrophysiology experiments (e.g., patch clamp) in synthetic lipid bilayer systems [Gilles, A.; Barboiu, M., Highly selective artificial K⁺ channels: an example of selectivity-induced transmembrane potential. *J. Am. Chem. Soc.* 138, 426-432 (2016); Debnath, M.; Chakraborty, S.; Kumar, Y. P.; Chaudhuri, R.; Jana, B.; Dash, J., Ionophore constructed from non-covalent assembly of a G-quadruplex and liponucleoside transports K⁺-ion across biological membranes. *Nat. commun.* 11, 1-12 (2020); Montenegro, J.; Ghadiri, M. R.; Granja, J. R., Ion channel models based on self-assembling cyclic peptide nanotubes. *Acc. Chem. Res.* 46, 2955-2965 (2013)]. In the manuscript, we present a set of experimental data and analyses that collectively support that **TBP2** indeed functions as an ion channel.

Fluorescence Microscopy and Co-localization: The fluorescence microscopy images (Figure 2J) demonstrated that **TBP2** embedded within lipid membrane as indicated by the co-localization of **TBP2** with the membrane staining dye Nile red. This suggests that **TBP2** binds to the vesicular membrane.

CF Release Assay: The CF release assay (Figure 2I, S2-SI) was employed to investigate whether **TBPs** can disrupt the lipid membrane. The data revealed that **TBP2** conserved membrane integrity while **TBP1** formed larger openings, releasing the CF dye (<10 Å in size) at significantly higher percentage (~52%) into the extravesicular solution. The CF data correlates with the fluorescence microscopy data, indicating **TBP2** could possess the ability to form channels.

HPTS Assay: We utilized HPTS (pH-sensitive fluorescent dye) assays (Figure 2A-F) to show that **TBP2** could potentially transport metal ions such as Na⁺ and K⁺. The increase in HPTS fluorescence intensity indicated ion influx, and **TBP2's** EC₅₀ values for K⁺ and Na⁺ further confirmed its ion transport activity. The fluorescence intensity of HPTS increased due to formation of channels leading to H⁺ efflux following influx of cations e.g., Na⁺, K⁺, Rb⁺, Cs⁺, Li⁺ etc. **TBP2** showed the highest ion transport activity across the model lipid bilayer at 20 μM concentration for K⁺ (~70%) and Na⁺ (~65%) in HEPES-NaCl or HEPES-KCl buffer (pH 7.4). The Hill equation was used to determine the EC₅₀ values (concentration of molecules to achieve 50% ion transport activity) of the peptidomimetics. **TBP2** exhibited EC₅₀ values of 5.8 μM and 7 μM for K⁺ and Na⁺, respectively. The Hill equation used here is as follows:

$$I = 1/(1 + (EC_{50}/[Channel])^n)$$

I = relative fluorescence intensity, [Channel] = concentration of **TBP1** or **TBP2**, n = Hill co-efficient. The EC₅₀ values of **TBP2** for K⁺ and Na⁺ were determined to be 3.1 μM and 14 μM while **TBP1** showed EC₅₀ values of 43 μM and 56 μM for K⁺, and Na⁺, respectively (Internal buffer: HEPES NaCl, pH 6.4) (Figure 2 and see the SI, Figure S1). When the internal buffer was substituted with KCl (10 mM HEPES 100 mM KCl, pH 6.4), **TBP2** displayed EC₅₀ values of 5.8 μM and 7 μM for K⁺ and Na⁺, respectively. Thus, HPTS studies revealed that **TBP2** can form channels, enabling efficient transportation of ions while **TBP1** did not show channel-forming activities.

Lucigenin Assay: In the revised manuscript, we employed a lucigenin assay (Figure 2G) to observe Cl^- transport activity of **TBP2**. The fluorescence intensity of lucigenin did not change significantly compared to blank with incremental addition of **TBP2**; suggesting no Cl^- flux across the lipid membrane.

Safranin O Assay: The safranin O assay (Figure 2H) was also conducted to investigate **TBP2**-dependent membrane polarization using GUVs. The data revealed that **TBP2** modulates membrane polarization in the presence of NaCl and KCl buffer solutions. For membrane polarization experiments, GUVs were prepared using 9:1 mixture of 10 mM EYPC and cholesterol via electroformation technique. In a fluorescence cuvette, 25 μL of GUV solution was suspended in 475 μL 10 mM HEPES, pH 6.4 containing the corresponding salt 100 mM NaCl or KCl. Next, safranin O was added at a final concentration of 1 μM . The fluorescence (with or without **TBP2**) was monitored for 600 sec at an excitation of 522 nm and emission of 581 nm (Figure 2H). The data showed that fluorescence intensity of safranin O significantly decreased with increasing concentration of ligand, suggesting **TBP2** modulated membrane polarization in the presence of NaCl (external) and KCl (internal) buffer solution.

Figure 2. G) Lucigenin assay of **TBP2** for Cl^- transport. Change in fluorescence intensity as a function of time in 225 mM NaNO_3 buffer. (H) Safranin O assay for membrane polarization in the presence of HEPES-NaCl and KCl buffers. Change in fluorescence of safranin O as a function of time (with or without **TBP2**).

Patch Clamp Experiments: We performed patch clamp experiments (Figure 3) using planar lipid bilayers, which show distinctive channel openings and closings upon addition of **TBP2**, confirming its channel-forming behavior for both Na^+ and K^+ .

Patch clamp experiments using planar lipid bilayers provided further insight into real-time channel formation behaviour of thiazolyl peptidomimetic **TBP2** (Figure 3). The addition of **TBP2** to the planar bilayer resulted in distinctive channel openings and closings at +80 mV and -80 mV, confirming the formation of channels across planar lipid bilayer membrane in the presence of both Na^+ and K^+ (Figure 3). I-V plot of **TBP2** shows an ohmic-linear relationship between current vs. voltage (Figure 3G). In the presence of 1 M NaCl at both *cis* and *trans* side, **TBP2** displayed multiple channel openings to transport Na^+ at -80 mV (Figure 3A). It also exhibited multiple square-top behavior at the holding potential of +80 mV in the presence of either NaCl or KCl. It formed multiple channel openings that were stable for longer duration (~ 2 sec) in the presence of Na^+ whereas it transported K^+ by forming channels that fluctuates more quickly between the closed and open conformations (Figure 3). **TBP2** efficiently transported Na^+ and K^+ across planar lipid bilayer membrane with high conductance. The average conductance values for transporting Na^+ and K^+ were measured to be ~ 0.56 nS and ~ 0.68 nS in the presence of 1 M NaCl and KCl, respectively. However, the current behaviour was also examined in the presence of CsCl, LiCl, RbCl buffers to illustrate the voltage dependent gating characteristics of **TBP2** for other monovalent cations (e.g., Cs^+ , Li^+ , Rb^+) besides Na^+ or K^+ . I-V curve suggests that **TBP2** possessed comparatively lower conductance values for other metal ions (e.g., Cs^+ : 0.29 nS, Li^+ : 0.08, Rb^+ : 0.15 nS) than Na^+ or K^+ in relation to applied positive or negative voltages (Figure 3).

Molecular Dynamics Simulation: Molecular dynamics simulations were employed to study the supramolecular arrangement of **TBP2** and its mechanism of ion channel formation. The simulation results suggested that **TBP2** can spontaneously transport Na^+ and K^+ ions through the channel, further supporting its ionophoric capabilities.

The molecular dynamics simulation illustrated the supramolecular arrangement of **TBP2** and provided mechanistic insights of ion channel formation in the lipid membrane. By performing molecular dynamics study, we tracked the number of Na^+ , K^+ or Cl^- present within the ion channel. Initially, no cations or anions were present within the channel. However, as shown in Figure 4, after only few picoseconds of simulations, cations entered the channel, and the number of cations continuously varied throughout the entire duration. We observed one or two Na^+ and K^+ ions to be present inside the channel for most of the time, while three cations were present in fewer time instants. An incessant change in the number of cations in the ion channel validated the exit of old ions as well as the entry of new cations, thereby suggesting spontaneous ion passage through the channel.

The average interaction energy of a single cation with the whole ion channel was computed to be -23.8 kcal/mol, and -14.7 kcal/mol for Na^+ and K^+ , respectively; which followed the same trend as obtained from our quantum chemistry calculations. Clearly, a Na^+ was more stable compared to a potassium ion within the channel.

In coherence with the DFT results, we observed only one chloride ion to be present within the stacked **TBP2** molecules in few time instants while in most of the time instants, no chloride ion entered the cavity (Figure 4C). Therefore, certainly the stacked **TBP2** molecular cavity had an inclination toward the binding and passage of cations of suitable size.

Altogether, the experimental and molecular dynamics simulation study suggest that **TBP2** could act as ionophore to transport ions across lipid membrane.

Figure 4. (C) Number of ions within the channel for the 500 ns production simulations for the channel-embedded lipid membrane in 0.15 M NaCl and KCl medium. (D) Free energy profiles (in kcal/mol) in terms of potential of mean forces (PMF's) for the passage of Na^+ , K^+ , and Cl^- ions through the ion-channel at 310 K, calculated employing the adaptive biasing force (ABF) module implemented in NAMD 2.12.

9) Why did the ion conductance profiles in Figures 3A and 3C or Figures 3B and 3D differ significantly from each other, while the I-V plots in Figure 3G show the linear profiles? From these results, it is difficult to conclude that the ion channels are formed in the membrane.

-The differences in ion conductance profiles at different voltages and ion types, combined with the linear I-V curve and the absence of currents without **TBP2**, indicate that **TBP2** possessed distinct voltage-

dependent gating mechanisms. The ion channels formed by **TBP2** responded differently to positive and negative voltages, exhibiting multiple conductance states and dynamic gating behavior.

Voltage-Dependent Gating Mechanisms: Different positive and negative voltages were applied (e.g., -80 to +80 mV) to measure the current across **TBP2** incorporated GUV membrane in the presence of MCl buffers (M = K⁺, Na⁺, Li⁺, Cs⁺, Rb⁺). The data suggested that **TBP2** exhibited different gating mechanisms in response to positive and negative voltages. At -80 mV in the presence of NaCl buffers, **TBP2** displayed multiple open states (Figure 3A), indicating that the ion channels were transitioning between different conductance states at negative voltages. In contrast, at +80 mV, **TBP2** exhibited square top behavior in the presence of NaCl buffers, suggesting that the ion channels might be reaching a fully open state more consistently at positive voltages. Similarly, when KCl buffers were present, **TBP2** transported K⁺ by forming channels that fluctuated rapidly between closed and open conformations at -80 mV. This rapid fluctuation implies dynamic gating mechanisms that respond differently to specific ion types and voltages.

Current vs. Count Histogram and Conductance Determination: Current vs. Count histogram in Figure 3 were plotted based on the current behavior of **TBP2** at specific voltages and the mean current with highest counts were considered to determine the conductance and I-V curve. The linear I-V curve obtained by plotting current values against different applied voltages suggests that there is a linear relationship between current and voltage within the range tested.

Evidence of Ion Channel Insertion: The absence of currents in the absence of **TBP2** or without applied voltages in the **TBP2**-incorporated GUVs strongly suggests that the ion channels are indeed responsible for the observed ion conductance. The ion channels are successfully inserted into the membrane and are contributing to the conductance measured.

[Chui, J. K.; Fyles, T. M., Ionic conductance of synthetic channels: analysis, lessons, and recommendations. *Chem. Soc. Rev.* 41, 148-175 (2012)].

10) Does Figure 4A show the initial structure of the ion channel formed by **TBP2** in the simulation? As a result of MD simulation, the alignment of **TBP2** molecules seems to be too regular. The authors should provide a movie of the MD simulation.

We thank the reviewer for this important concern. Indeed, Figure 4A showed the initial structure of the ion channel formed by **TBP2** molecules, and after starting the simulations they slightly changed their position during equilibration. However, to avoid large deviation, during the ion transport simulations, we harmonically constrained the center of mass of the whole **TBP2** supramolecular assembly, so that the assembly can reorient themselves if required, but the center of mass did not move, thereby restricting any lateral movement of the assembly as a whole. In the revised manuscript, we have provided a model simulation movie, which shows how a single Na⁺ ion is transported through the ion channel. The movie is provided with the manuscript as an separate attachment.

11) What about the effect of **TBP2**-induced ion permeation on the cell viability?

The authors thank the reviewer for raising concern about the intracellular effect of **TBP2** due to ion permeation via the cell membrane, if **TBP2** mediates the cancer cell death by ionic imbalance in the cells. We have performed fluorescence experiments to examine the **TBP2** induced changes in the cellular concentration of Na⁺ or K⁺ with a known sodium or potassium binding fluorescent dye e.g., Sodium-binding benzofuran isophthalate acetoxymethyl ester (SBFI-AM, Na⁺ probe) and Potassium-binding benzofuran isophthalate acetoxymethyl ester (PBFI-AM, K⁺ probe). HeLa or A549 cells were incubated with SBFI-AM or PBFI-AM (10 μM) for 2 h at culture conditions following treatment with **TBP2** with low to higher concentration (0-8 μM) for 1.5 h. The intracellular fluorescence measurements revealed that **TBP2** (8 μM) could significantly increase the concentration of Na⁺ up to ~ 37% and marginally decrease the cytosolic K⁺ concentration ~ 15% in HeLa cells (Figure 5E). However, ~19% increase in Na⁺ and ~10% decrease in K⁺ concentration were observed in **TBP2** embedded A549 cells compared to control. These results could be rationalized by different Na⁺ or K⁺ distributions across the cell membranes as the intracellular Na⁺ concentrations (~12 mM) are much lower than the extracellular Na⁺ (~145 mM), whereas the extracellular potassium concentration (~4 mM) are much lower than its intracellular counterpart (~150 mM) in physiological conditions. Hence, the data indicated that **TBP2** induced

intracellular increase in Na⁺ could disrupt the ionic balance and synergize the formation and stabilization of G-quadruplex structures following accumulation of **TBP2** after long duration in cancer cells nuclei.

Figure 5E. **TBP2** dependent intracellular fluorescence intensity measurements of Sodium-binding benzofuran isophthalate acetoxymethyl ester (SBFI-AM, Na⁺ probe) and Potassium-binding benzofuran isophthalate acetoxymethyl ester (PBFI-AM, K⁺ probe) in i) HeLa and ii) A549 cells.

It is noteworthy to state that the biophysical and biological experiments indicated that **TBP2** could embed upon membrane instantly to form the synthetic ion channels via lipid membrane while it binds to G-quadruplexes inside the cancer cell nuclei in a time-dependent manner (i.e., it kills the cancer cells after 24 h of treatment with an IC₅₀ value of ~8.2 μM while it does not affect the normal kidney epithelial cells after similar time interval). These properties make **TBP2** a unique synthetic ion transporter to mediate cancer cell death via G4 dependent mechanism.

12) Figure 5A does not show the localization of **TBP2** molecules to the nuclei, but indicates that they are present in both the nuclei and cytoplasm.

The authors thank the reviewer for this comment. As observed in Figure 5A, **TBP2** (blue) efficiently penetrates the cell membrane and accumulates inside the HeLa cell nuclei (nucleus is indicated with white line). The images also suggested that the concentration of **TBP2** was significantly higher inside the cellular nuclei compared to cytoplasm of HeLa cells. The number of G4 binding antibody (BG4) foci (as seen as punctuated dots) was significantly increased in **TBP2** treated cells compared to the control cells or **TBP1** treated cells (Figure 5A, B), suggesting the potential of **TBP2** to stabilize G4s in cellular system.

Figure 5A. Confocal images of HeLa cells (fixed) stained with **TBP2** (blue) and BG4 (red); scale bars represent 10 μm . Nucleus of each cell is indicated with white borderline.

13) The difference in the dissociation constants between *c-MYC22 G4*, *c-KIT1*, *h-TELO*, *ds26 DNA* and **TBP2** is less than 10-fold and not very large, whereas the difference in the fluorescence intensity change during titration between *c-MYC22 G4* and others is significant. What is the reason for this?

The titration experiments were carried out with the ligand **TBP2** at 2.5 μM concentration in the presence of pre-annealed quadruplexes and *ds26 DNA* in 60 mM potassium cacodylate buffer (pH 7.4). Fluorescence of **TBP2** was measured between 340 nm and 550 nm using the excitation wavelength of 330 nm in degassed 60 mM KCaco buffer (pH 7.4). The specific concentrations of pre-annealed DNAs were added into the **TBP2** solutions and incubated for 2 min before fluorescence measurement. The binding affinity or the apparent K_d was calculated by the Hill-1 equation using OriginPro 2016 software. The equation is as follows:

$$F = F_0 + \frac{(F_{max} - F_0) [DNA]}{K_d + [DNA]}$$

Where F signifies for fluorescence intensity, F_{max} for maximum fluorescence intensity, F_0 for fluorescence intensity in the absence of DNA and K_d for dissociation constant.

	TBP2	
DNA	K_d	F/F_0
c-KIT1	9.5	1.56
c-KIT2	9.2	1.62
c-MYC	1.6	7.82
BCL-2	10.2	1.45
h-TELO	8.2	1.82
ds26	8.2	1.82

Table R1. The fold difference (F/F_0) and K_d values (calculated by Hill1 equation) of **TBP2** after titration with different G-quadruplexes and *dsDNA* in potassium cacodylate buffer.

The fluorescence titration of **TBP2** with DNA quadruplexes and *ds26* DNA demonstrated that the fluorescence intensity of **TBP2** changed up to ~8 fold after addition of *c-MYC22* quadruplex DNA while ~1.6 fold for *c-KIT1* and *c-KIT2*, ~1.4 fold for *BCL-2*, and ~1.8 fold for *h-TELO* and *ds26* DNA (Table S1). Hill1 equation was used to determine the binding affinity where the degree of change with each addition of DNAs during titration is also considered in K_d evaluation (Lakowicz, J. R., *Principles of fluorescence spectroscopy*. Springer science & business media: 2013). Thus, the binding affinity derived from fluorescence measurement significantly differ in case of *c-MYC22* quadruplex DNA in relation to other investigated quadruplexes and *dsDNA*.

14) It is necessary to explain why **TBP2** shows a particularly strong interaction with *c-MYC22* G4 than others.

(also see Response 3, Referee 2)

-Based on the biophysical results and molecular dynamics simulation of **TBP2** with *c-MYC22*, the explanation for specific interaction of **TBP2** with *c-MYC22* is provided in the manuscript:

We performed the NMR and molecular dynamics simulation study to understand the insights into the interaction of **TBP2** with *c-MYC*. The interaction of **TBP2** with *c-MYC22* was investigated by recording 1D 1H NMR spectra at different [**TBP2**]:[DNA] ratios (Figure 6E, F, see the Supporting Information, Figure S11). The titrations were performed in 25 mM Tris·HCl buffer containing 100 mM KCl. The *c-MYC22* DNA formed a definite G-quadruplex structure that was described by Yang et al. (PDB: 1XAV). The DMSO effect on the *c-MYC22* G-quadruplex was previously discussed (Kumar, Y. P.; Bhowmik, S.; Das, R. N.; Bessi, I.; Paladhi, S.; Ghosh, R.; Schwalbe, H.; Dash, J., A Fluorescent Guanosine Dinucleoside as a Selective Switch-On Sensor for c-myc G-Quadruplex DNA with Potent Anticancer Activities. *Chem. Eur. J.* 19, 11502-11506 (2013)) and the assignment for the target DNA was followed from Yang et. al. (Ambrus, A.; Chen, D.; Dai, J.; Jones, R. A.; Yang, D., Solution structure of the biologically relevant G-quadruplex element in the human *c-MYC* promoter. Implications for G-quadruplex stabilization. *Biochemistry* 44, 2048-2058 (2005)). Clear changes in the 1D 1H spectrum upon addition of the ligand to the G-quadruplex formed by the sequence *c-MYC22* were detected. The addition of 0.25 equivalents **TBP2** led to general line broadening of the imino signals (Figure 6E). However, further addition revealed stronger effects for the imino signals of G4, G9, G10, G13 and G17. For the imino signal of G6, a second signal for the bound state seemed to appear that was present in a 1:1 ratio at a [ligand:DNA] molar ratio of 1. In the aromatic region, the signals of A3, G2 or G4, G5, G13, T20, A21H2, A22, A22H2 and some signals of the overlapping region were mostly affected by ligand-binding (see the Figure S11, SI). The imino and aromatic signals showing the largest changes upon addition of **TBP2** are marked in red and with spheres on the NMR-structure of *c-MYC22* (pdb-code 1XAV). The data revealed that the strongest effect was observed for nucleotides on the lower and upper

tetrad as well as both capping structures and G5 and G9 from the middle tetrad (Figure 6E, F, Supplementary Information, Figure S11). Thus, the NMR data indicated that the interaction of **TBP2** took place on the bottom and top of the G-quadruplex and there is probably a direct interaction with G6 as well.

Figure R1. 1D ^1H NMR spectra of the (A) imino and (B) aromatic region of *c-MYC22* in 25 mM Tris-HCl buffer containing 100 mM KCl at different [**TBP2**:DNA] molar ratios.

We further employed the classical MD simulations to understand and compare the interaction profile of **TBP2** with two topologically distinct G4s *c-MYC22* (PDB: 1XAV) and *h-TELO* (PDB: 1KF1). For each of the G4s, two sets of MD simulations were carried out by placing the **TBP2** molecule at a distance of ~ 12 Å from the upper and lower tetrad of the quadruplex. Figure 7(A) displays the top and side views of the final composite structures for both the G4s in both sets of simulations. The simulation results depicted that **TBP2** molecule binds to the upper and bottom tetrad moiety of both the G4s; which is in coherence with NMR results. Further analyses showed that the A2, G4, G5, G13, G17, T20, A21, and A22 residues of *c-MYC* while for *h-TELO*, the G2, G3, G4, T5, T11, G15, T17, G21, and G22 residues interact with **TBP2**. The H-bond calculation (Figure 7B) indicated that 2-3 H-bonds were formed between *c-MYC* and **TBP**, while *h-TELO* possessed only one H-bond with the molecule. However, the residual connections between the G4s and **TBP2** originated from the van der Waals and long-range electrostatic interactions. The free binding energies of **TBP2** were determined to be -11.5 kcal/mol, and -3.9 kcal/mol with *c-MYC* and *h-TELO* G4s, respectively, suggesting its favourable binding with *c-MYC* as observed in biophysical results. The simulation images (Figure 7A) indicated that **TBP2** fits on the top or bottom of *c-MYC*, thereby possesses higher binding energy, while the fitting cavity of *h-TELO* could not adequately contain the **TBP2** molecule, resulting in a weaker interaction.

Figure 7. Molecular dynamics simulation of **TBP2** (a) Top and side views of the final structures of **TBP2**, interacting with the top and bottom tetrad of *c-MYC* (1XAV) and *h-TELO* (1KF1) G-quadruplexes, as observed from two MD simulation trajectories. (b) Number of H-bonds between **TBP2** and G-quadruplexes. (c) Free energies of binding between **TBP2** and the two G-quadruplexes. Since the **TBP2** molecule undergoes adsorption at different distances from the two G-quadruplexes, therefore, for a direct comparison, the reaction coordinate is represented as $\delta = d - d_0$, where, d and d_0 are the distances at any time instant and the initial distance of adsorption, respectively.

Reviewer #2 (Remarks to the Author):

In this manuscript Dash and coworkers describe the synthesis and the biophysical characterisation of a peptide-mimic (**TBP2**) and its cellular properties as a G4-binder. The manuscript is effectively split in 2 parts. In the first part, the authors described the biophysical properties of **TBP2** in terms of aggregation and as a peptide-like ion channel in artificial membranes, which could lead to K⁺ transport in artificial cells. This part of the manuscript is very well described and characterised, with a series of biophysical techniques used to characterise the ion channel properties of **TBP2**. In the second part of the manuscript the authors apply **TBP2** in cells and study their effect on G4-stabilisation. There is more confusion here and is hard to understand if the authors are implying that **TBP2** binds as a single entity to nuclear G4s or if the molecule induces K⁺ increases in cells, thus causing formation of G4s. The authors then go back to biophysics to show that **TBP2** has a preferential binding to MYC G4 and speculate this is the reason why they observe gene-expression downregulation using a luciferase assay. Overall, my main issue is that these 2 part of the manuscript they are very disjoint and is hard to understand how they are linked, they sound more like two separate manuscripts. Furthermore, the second part is very speculative and not enough evidence are provided to justify the mechanisms suggested in Figure 5.

The authors thank the reviewer for careful evaluation of our work and providing useful suggestions to more evidently link the **TBP2** induced ion channel with intracellular G4-mediated cell death. Based on the comments from both the reviewers, we have performed necessary additional experiments and literature survey to further conclude our work and incorporate them in the main manuscript or supplementary information.

Scheme 1. Diagrammatic representation of **TBP2** gated ion channel formation and cell death in time-dependent manner.

Below are my main concerns:

1) There is no link between the biophysical properties of the **TBP2** as ion channel and its cellular activity. The authors imply in figure 5 that there is an increase in K⁺ levels but they provide no evidence to support this. How the biophysical characterisation of **TBP2** as ion channel is relevant for its G4-binding property as a monomer is not clear at all.

The authors thank the reviewer for raising concern about the biophysical and biological properties of **TBP2** as ion channels and G4 binder. The biophysical data and fluorescence imaging studies using LUVs, GUVs or live cells demonstrated that **TBP2** instantly embedded upon vesicular or cell membrane to act as ion channels. The imaging and molecular dynamics simulation studies depicted the supramolecular self-assembling property of **TBP2** in the lipid membrane where the molecules tend to stack upon each

other to form channels. However, intracellular studies indicated that large concentration of **TBP2** can be found inside the cellular nuclei after 24h which could be due to amphipathic nature of the molecule.

We performed the cell based fluorescence experiments to examine the **TBP2** induced changes in the cellular concentration of Na⁺ or K⁺ with a known sodium or potassium binding fluorescent dye e.g., Sodium-binding benzofuran isophthalate acetoxymethyl ester (SBFI-AM, Na⁺ probe) and Potassium-binding benzofuran isophthalate acetoxymethyl ester (PBFI-AM, K⁺ probe). HeLa or A549 cells were incubated with SBFI-AM or PBFI-AM (10 μM) for 2 h at culture conditions following treatment with **TBP2** with low to higher concentration (0-8 μM) for 1.5 h. The intracellular fluorescence measurements revealed that **TBP2** (8 μM) could significantly increase the concentration of Na⁺ up to ~ 37% and marginally decrease the cytosolic K⁺ concentration ~ 15% in HeLa cells (Figure 5E). However, ~19% increase in Na⁺ and ~10% decrease in K⁺ concentration were observed in **TBP2** embedded A549 cells compared to control. These results could be rationalized by different Na⁺ or K⁺ distributions across the cell membranes as the intracellular Na⁺ concentrations (~12 mM) are much lower than the extracellular Na⁺ (~145 mM), whereas the extracellular potassium concentration (~4 mM) are much lower than its intracellular counterpart (~150 mM) in physiological conditions. Thus, the data suggested that significant intracellular increase of monovalent cation Na⁺ might disrupt the ionic balance, thereby synergistically contributing to the formation and widespread stabilization of G-quadruplex structures. This effect is likely facilitated by the substantial accumulation of **TBP2** within the nuclei of cancer cells after an extended incubation period.

Figure 5E. **TBP2** dependent intracellular fluorescence intensity measurements of Sodium-binding benzofuran isophthalate acetoxymethyl ester (SBFI-AM, Na⁺ probe) and Potassium-binding benzofuran isophthalate acetoxymethyl ester (PBFI-AM, K⁺ probe) in i) HeLa and ii) A549 cells. iii) Schematic illustration of **TBP2**-G4 interaction.

2) The confocal microscopy clearly show wide **TBP2** distribution in the nuclei of human cancer cells (Figure 5), so even if biophysically there is some selectivity for binding *MYC* G4, this is clearly not relevant under the conditions used for cellular experiments, as revealed by BG4 co-localisation. So how the *MYC* selectivity data is relevant is hard to appreciate.

The authors agree with the reviewer that the immunofluorescence study using BG4 antibody illustrates the binding potential of **TBP2** with the G4 structures while it does not clarify the selectivity profile in cellular system. We acknowledge the reviewer's suggestion in point 3 to perform qRT-PCR to examine

TBP2 mediated downregulation of other G4 containing oncogenes besides *c-MYC* (please refer to response 3).

3) On a similar note to point 2 the authors use a luciferase assay with *MYC* promoter to assess the ability of **TBP2** to bind *MYC* G4, which is fine, but clearly cannot be used as an indicator of selective binding for this particular G4 in cells. RT-qPCR data should be collected looking also at other promoters containing G4s (i.e. *KIT*, *SRC* etc) and show preferential down regulation of *MYC*, if the authors want to claim selectivity.

The luciferase reporter assay was performed to evaluate whether **TBP2** could possess the potential to bind to the G4 harboring promoter region of *c-MYC* oncogene. Consequently, the luciferase assay only suggested that **TBP2** efficiently interacted with the promoter to repress *c-MYC* transcription in HeLa cells.

In the revised manuscript, qRT-PCR experiments involving other G4-forming oncogenes like *c-KIT*, *BCL-2*, in addition to *c-MYC*, were carried out to illustrate the selectivity profile of **TBP2** at the mRNA level. The results of the real-time PCR analysis showed that **TBP2** could decrease *c-KIT* expression by approximately 6% at 4 μM and around 59% at 8 μM . Similarly, the expression of *BCL-2* was reduced up to approximately 20% at the highest concentration after 24 hours, compared to control HeLa cells (as shown in Figure 5C). Notably, **TBP2** exhibited the ability to downregulate *c-MYC* mRNA levels by ~44% at 4 μM and a substantial 90% at 8 μM . As such, these qRT-PCR findings were consistent with the outcomes of other biophysical and biological studies, providing supporting evidence for the potential of the peptide mimic **TBP2** to selectively downregulate the expression of the *c-MYC* oncogene within cancer cells.

Figure 5. (C) qRT-PCR analysis for transcriptional regulation of *c-MYC*, *BCL-2*, *c-KIT* oncogene after treatment with **TBP2** in HeLa cells for 24 h. Quantification was done in terms of fold change by double delta C_T method using 18s rRNA as a housekeeping gene. Fold change of ligand treated relative gene expression is normalized with control. Three biological replicates were employed for the quantifications. Error bars represent mean \pm SD. * $P < 0.05$ and ** $P < 0.001$ (Student's t test), versus control HeLa cells.

4) The authors show BG4 foci increase upon treatment with **TBP2**, so again their evidence suggest broad G4-stabilisation. If the authors want to speculate this is due to the localisation of the molecule in the plasma membrane of the HeLa cells then they should be able to quantify K^+ concentrations increasing after treatment with the molecule. Otherwise is hard to distinguish the effect of stabilisation of the monomer vs the ion channel effect (if any).

The authors appreciate the reviewer's suggestion. As illustrated in Figure 2J, **TBP2** promptly integrated into the membrane upon treatment, and its accumulation within the cellular nuclei became evident after extended incubation, such as 24 hours. This accumulation could have potentially contributed to intracellular G4 stabilization (increase in BG4 foci), eventually leading to cell death. Furthermore, we conducted measurements of intracellular Na^+ and K^+ concentrations following a 2-hour treatment with

TBP2 (please refer to response 1 for details). This additional investigation provided insights into the potential changes in ion concentrations induced by **TBP2**. By considering these factors, we showed the effects of monomer stabilization versus ion channel activity.

5) Overall this is a potentially interesting manuscript with some interesting preliminary evidence. However, currently the data reported are scattered and disjoint not really allowing the authors to support their conclusions. I would recommend to either split this manuscript into 2 smaller articles or to accumulate more evidence to fully support the author's speculation on the cellular mechanism of action.

The authors thank the reviewer for the comments and interest in our work. The manuscript has been revised diligently, incorporating valuable suggestions from both reviewers. Additional experiments were performed as suggested to establish our conclusions.

In summary, we designed a peptide mimic demonstrating the ability to form vesicular and nanofibrillar non-covalent frameworks in response to specific microenvironments. **TBP2** was established as capable of creating transmembrane channels across lipid membranes, influencing intracellular ion concentrations. Moreover, it possesses structural features that enable it to bind to non-canonical DNA secondary structures. It could also form supramolecular nanostructures, giving rise to distinctive channel behaviour in the presence of alkali metal ions, such as Na⁺ and K⁺. Notably, **TBP2** rapidly localized in model lipid bilayers and the plasma membrane to facilitate ion transport. Over an extended period (e.g., 24 hours), it exhibited time-dependent accumulation within the nuclei of HeLa cells, evident through enhanced fluorescence.

TBP2's versatility positions it as a promising molecular scaffold for exploring biological systems. Channel formation by **TBP2** results in higher intracellular levels of monovalent cations like Na⁺, crucial component of DNA G4 structures. The interplay of time-dependent membrane or nuclear localization, followed by increased intracellular cation concentration and direct G4 interaction, synergistically promote G4 formation and stability, ultimately triggering cancer cell death. This class of synthetic peptide nanostructures not only provides insights into ion channels but also introduces a novel approach for creating therapeutic artificial transporters.

Reviewer #1 (Remarks to the Author):

The authors sincerely addressed the questions raised by the reviewers and revised the manuscript with additional experimental results to clarify the properties of TBP2 and its function in cellular systems. However, it is still unclear how the dual function of TBP2, namely, the ion transport capability and regulation of gene function, is correlated with each other. If these functions work independently to trigger the cancer cell death, it is difficult to find the advantage of the present system. In fact, the increase of the sodium concentration in the cell systems due to TBP2 is not so large, and it is difficult to find how large the ion transport ability of TBP2 affects on the stabilization of the G-quadruplex of the specific genes. In addition, while the authors claim the formation of ion channels in the cell membrane by TBP2, they also claim that TBP2 penetrates into the cell nuclei. The ability of TBP2 to escape from the membrane suggests the possibility that TBP2 may also act as a mobile carrier although it forms ion channels. It is also unclear how TBP2 is delivered to nuclei, e.g. with or without nanofiber formation, etc. Although the manuscript shows significant improvement over the previous version in terms of characterization of TBP2 molecules and evaluation of their ion transport properties, it is still considered that the overall argumentation of this manuscript is insufficient to meet the criteria of Nature Communications. This reviewer recommends that the content be split into two parts and submitted to more specialized journals.

Reviewer #2 (Remarks to the Author):

The authors have addressed my main concerns during the revision stage and I am therefore supportive of publication in Nat Commun of the amended manuscript.

Reviewer #3 (Remarks to the Author):

Review report on "Expanding the landscape of synthetic ion transporters: A Non-B DNA binding Supramolecular peptide channel Alters Cellular Functions"

The manuscript presents a study on a specific ligand that has an ability of self-assembling into a nanostructure. The authors tried to prove that this nanostructure may have a structure of ion channel or transporter. To be more specific, the authors propose that such a channel has a structure shown in figure 1(ii)C or figure 4(A). To provide evidence for this proposed model, the authors perform a series of experiments including TEM image, various assays, patch clamp, and perform MD simulations to show that ions (K⁺ and Na⁺) can permeate a model channel.

However, to this specific claim that the assemble may have a channel-like structure as proposed, I don't find it 100% convincing. The model suggests an assembly of 6 TBP2 spanning a membrane, implying a well structure, so perhaps, cooperativity = 6, while the Hill coefficient of TBP2 for K⁺ is only 1.64, implying the cooperativity of more than 1 TBP2 but less than 2. Furthermore, the evidence shown in Figure 2J for TBP2 may suggest a likely unstructured of nano-assembly of TBP2 across the membrane. I think the model was biased by the DFT optimization performed on the structure, which was more or less in a channel-like structure. Figure 3(A-D) might not exhibit some kind of channel-like functions at all. Yes, ions are permeating the membrane, but the currents appear very stochastic, burst-like. Such a behavior might indicate some ruptures in the structure of the assembly that enable ions to permeate. The currents' behaviors appear to depend on voltage polarity as well. This might suggest asymmetric structure of the assembly, while the proposed model has a cylindrical structure.

A more convincing piece of evidence would be that a set of TBP2 in random positions in a lipid membrane is simulated to form a channel-like structure.

Some paragraphs do not refer their statements to specific figures, which are quite dense with many

sub-figures. It causes some difficulties in understanding the data and how to relate it to the statements in the text.

The authors might refer to some DFT calculations of Na⁺ and K⁺ for their interactions with different environments, such as, DOI: 10.1021/jp510560k.

Gramicidin A (gA) channel might be an interesting counterpart to compare the proposed model with (which I doubt it would be like as proposed). For gA, a typical barrier for K⁺ and Na⁺ 4-8 kcal/mol, and conductance is less than 24 pS (see 10.1021/acs.jctc.0c00968), and it is of course quite a stable channel.

For studies of cell deaths, I am not convinced that TBP2 would only kill cancer cells. The authors should present clear evidence for the action of TBP2 for normal cells versus cancer cells. From the current evidence, I could argue that TBP2 may be highly toxic. The argument that "TBP2 possessed the ability to stabilize G4 structures in cancer cells", thus "induce cancer cell deaths" is highly speculative. There are many possibilities that a ligand can kill a cell. It may stop signaling pathways or may cause damages to mitochondria by simply blocking VDAC1 channels, or interfere with transcription processes. Killing a cell is easy, but killing only cancerous cells is actually a very hard problem. The author must demonstrate such a specificity to make such a claim.

I feel that the manuscript has two separate stories, one is to establish a channel-like structure of TBP2 assemblies, while the other is the interaction of TBP2 and G4 and its implication for treating cancer. Both stories are quite interesting and warrant publications somewhere else, even though the writing of this MS makes it hard to follow and understandable to a wider audience.

Point-by-point response to referees

Reviewer #1 (Remarks to the Author):

The authors sincerely addressed the questions raised by the reviewers and revised the manuscript with additional experimental results to clarify the properties of TBP2 and its function in cellular systems. However, it is still unclear how the dual function of TBP2, namely, the ion transport capability and regulation of gene function, is correlated with each other. If these functions work independently to trigger the cancer cell death, it is difficult to find the advantage of the present system. In fact, the increase of the sodium concentration in the cell systems due to TBP2 is not so large, and it is difficult to find how large the ion transport ability of TBP2 affects on the stabilization of the G-quadruplex of the specific genes. In addition, while the authors claim the formation of ion channels in the cell membrane by TBP2, they also claim that TBP2 penetrates into the cell nuclei. The ability of TBP2 to escape from the membrane suggests the possibility that TBP2 may also act as a mobile carrier although it forms ion channels. It is also unclear how TBP2 is delivered to nuclei, e.g. with or without nanofiber formation, etc. Although the manuscript shows significant improvement over the previous version in terms of characterization of TBP2 molecules and evaluation of their ion transport properties, it is still considered that the overall argumentation of this manuscript is insufficient to meet the criteria of Nature Communications.

-The authors thank the reviewer for appreciating the revised manuscript.

Scheme 1. Diagrammatic representation of **TBP2** gated ion channel formation and cell death in time-dependent manner.

- In the revised manuscript, we show that **TBP2** rapidly localizes in model lipid bilayers and the plasma membrane to facilitate ion transport, while it accumulates in the nuclei of HeLa cells after prolonged duration (e.g., 24 h) in a time-dependent manner. The formation of channels by **TBP2** leads to increase in intracellular concentration of monovalent cations like Na⁺, the essential components of DNA G4 structures. The interplay of time-dependent membrane or nuclear localization of **TBP2**, followed by its increased intracellular cation concentration and direct interaction with G4, synergistically promotes G4 formation and stabilization, triggering cancer cell death.

We conducted fluorescence experiments to examine the changes in intracellular Na⁺ or K⁺ concentrations induced by **TBP2** using specific fluorescent dyes e.g., sodium-binding benzofuran isophthalate acetoxymethyl ester (SBFI-AM, Na⁺ probe) and potassium-binding benzofuran isophthalate acetoxymethyl ester (PBFI-AM, K⁺ probe), respectively. HeLa or A549 cells were incubated with SBFI-AM or PBFI-AM (10 μM) for 2 h under standard culture conditions, followed by treatment with varying concentrations of **TBP2** (ranging from 0 to 8 μM) for 1.5 h. The intracellular fluorescence measurements revealed that **TBP2** (8 μM) could significantly increase the concentration of Na⁺ up to ~ 37% and marginally decrease the cytosolic K⁺ concentration ~ 15% in HeLa cells (Figure 5E). However,

~19% increase in Na⁺ and ~10% decrease in K⁺ concentration were observed in **TBP2** embedded A549 cells compared to control. These results could be rationalized by different Na⁺ or K⁺ distributions across the cell membranes as the intracellular Na⁺ concentrations (~12 mM) are much lower than the extracellular Na⁺ (~145 mM), whereas the extracellular potassium concentration (~4 mM) are much lower than its intracellular counterpart (~150 mM) in physiological conditions.

The growth inhibitory effect of **TBP1** and **TBP2** was assessed after a minimum of 24 h of treatment in both cancer and normal kidney epithelial cells. The cell viability study suggests that **TBP2** can induce cancer cell death [IC₅₀ values of ~8.2, ~10 and ~16 μM in cervical carcinoma (HeLa), leukemia (K562) and lung carcinoma (A549) cells, respectively], while normal kidney epithelial (NKE) cells remain unaffected up to 100 μM. However, **TBP1** did not induce cell death in either cancer or normal cells up to 100 μM as determined via the XTT based cell viability assay.

It is noteworthy to mention that, a 37% increase in Na⁺ concentration over a duration of 2 h can be considered significant in the cellular microenvironment [Park, S.-H.; Hwang, I.; McNaughton, D. A.; Kinross, A. J.; Howe, E. N.; He, Q.; Xiong, S.; Kilde, M. D.; Lynch, V. M.; Gale, P. A., Synthetic Na⁺/K⁺ exchangers promote apoptosis by disturbing cellular cation homeostasis. *Chem* **7**, 3325-3339 (2021)]. Thus, the data suggest that this intracellular increase in monovalent cation Na⁺ may disrupt the ionic balance, thereby synergistically contributing to the formation and widespread stabilization of G-quadruplex structures (monovalent cations, like Na⁺ are the basic structural component of G-quadruplexes) in cancer cells. This effect is observed in the cancer cells due to the possible accumulation of **TBP2** within the nuclei after an extended incubation period. The observed non-toxicity of **TBP2** towards normal cell lines may be attributed to the lower abundance of G4 structures in non-cancerous cells.

Figure 1. (ii)(A) TEM image of **TBP2** showing distribution of vesicles (indicated with purple arrow). B) The average size distribution (±SD) and relative abundance of **TBP2** formed vesicles as measured by DLS method.

TBP2 formed the nanofiber and few vesicle like structures which could be the result of strong aggregation of the aromatic surfaces of the molecules in the presence of cations (Figure 1(ii)). The self-assembled supramolecular nanofiber like structures thus played a key role in the formation of ion channels through lipid membrane. The supramolecular assembly embeds upon lipid membrane while the monomeric structures penetrates the cell membrane and accumulates in the nuclei following interaction with G4 structures. The supramolecular self-assembled ion channels facilitate transportation of Na⁺ or K⁺ through the lipid membrane and synergistically stabilize the G4s in cellular system along with **TBP2**. Besides that, colocalization study in cellular system revealed that **TBP2** could embed upon cell membrane after 10 minutes of compound treatment.

Figure S3. Membrane colocalization of **TBP2** with Nile Red in HeLa cells after 10 minutes of incubation. Scale bars represent 10 μm .

We believe the manuscript has been improved with additional suggested experiments and explanations to link the dual functionality of **TBP2** to act as both ion channels and intracellular G4 binder. **TBP2** rapidly localizes in model lipid bilayers and the plasma membrane to facilitate ion transport, while it accumulates in the nuclei of HeLa cells after prolonged duration (e.g., 24 h) in a time-dependent manner. The formation of channels by **TBP2** leads to increase in intracellular concentration of monovalent cations like Na^+ , the essential components of DNA G4 structures. The interplay of time-dependent membrane or nuclear localization, followed by increased intracellular cation concentration and direct G4 interaction of **TBP2**, synergistically promote G4 formation and stability, to trigger cancer cell death.

Reviewer #2 (Remarks to the Author):

The authors have addressed my main concerns during the revision stage and I am therefore supportive of publication in *Nat Commun* of the amended manuscript.

The authors thank the reviewer for appreciating the study and recommend the revised manuscript in *Nat Commun*.

Reviewer #3 (Remarks to the Author):

1. The manuscript presents a study on a specific ligand that has an ability of self-assembling into a nanostructure. The authors tried to prove that this nanostructure may have a structure of ion channel or transporter. To be more specific, the authors propose that such a channel has a structure shown in figure 1(ii)C or figure 4(A). To provide evidence for this proposed model, the authors perform a series of experiments including TEM image, various assays, patch clamp, and perform MD simulations to show that ions (K^+ and Na^+) can permeate a model channel.

However, to this specific claim that the assemble may have a channel-like structure as proposed, I don't find it 100% convincing. The model suggests an assembly of 6 TBP2 spanning a membrane, implying a well structure, so perhaps, cooperativity = 6, while the Hill coefficient of TBP2 for K^+ is only 1.64, implying the cooperativity of more than 1 TBP2 but less than 2. Furthermore, the evidence shown in Figure 2J for TBP2 may suggest a likely unstructured or nano-assembly of TBP2 across the membrane.

-The authors thank the reviewer for showing interest in our study and additional comments.

The fluorescence based HPTS assay was primarily performed to demonstrate the ion channel forming activity of thiazole peptide mimics **TBP1** and **TBP2**. To gain more insights into the ion transporting behaviour, Hill analysis was performed using the following equation:

$$I = 1/(1 + (EC_{50}/[Channel])^n)$$

I = relative fluorescence intensity, [Channel] = concentration of **TBP1** or **TBP2**, n = Hill co-efficient.

The Hill equation was used to determine the EC_{50} values (concentration of a molecule to achieve 50% ion transport activity) of the peptidomimetics. **TBP2** exhibited EC_{50} values of 5.8 μ M and 7 μ M for K^+ and Na^+ , respectively. Besides that, the Hill coefficient (n) assumes the possible co-operativity of a ligand to act as channels through the lipid membrane. Here, the Hill coefficient of **TBP2** for K^+ ($n_{K^+} = 1.64$) indicates that more than one active molecule (i.e., positive co-operativity) could participate to form supramolecular channel like structures to transport K^+ . [Gilles, A.; Barboiu, M., Highly selective artificial K^+ channels: an example of selectivity-induced transmembrane potential. *J. Am. Chem. Soc.* 138, 426-432 (2016)]. Based on the optimized structure of the compound, we assume that six molecules might participate (which is >1, indicating positive co-operativity) in the formation of supramolecular channel-like structures to span the lipid membrane [C. A. Schalley, *Analytical methods in supramolecular chemistry, Vol. 1*, John Wiley & Sons, 2012] (please refer to response 3).

2. Figure 3(A-D) might not exhibit some kind of channel-like functions at all. Yes, ions are permeating the membrane, but the currents appear very stochastic, burst-like. Such a behavior might indicate some ruptures in the structure of the assembly that enable ions to permeate. The currents' behaviors appear to depend on voltage polarity as well. This might suggest asymmetric structure of the assembly, while the proposed model has a cylindrical structure.

-The differences in ion conductance profiles at different voltages and ion types, combined with the linear I-V curve and the absence of currents without **TBP2**, indicate that **TBP2** possessed distinct voltage-dependent gating mechanisms. The ion channels formed by **TBP2** responded differently to positive and negative voltages, exhibiting multiple conductance states and dynamic gating behavior.

Figure R1. Current measurement in the presence of 1 M NaCl or 1 M KCl at 0 mV and +80 mV.

Different positive and negative voltages were applied (e.g., -80 to +80 mV) to measure the current across **TBP2** incorporated GUV membrane in the presence of MCl buffers (M= K^+ , Na^+ , Li^+ , Cs^+ , Rb^+). The data suggested that **TBP2** exhibited different gating mechanisms in response to positive and negative voltages. At -80 mV in the presence of NaCl buffers, **TBP2** displayed multiple open states (Figure 3A), indicating that the ion channels were transitioning between different conductance states at negative voltages. In contrast, at +80 mV, **TBP2** exhibited square top behavior in the presence of NaCl buffers, suggesting that the ion channels might be reaching a fully open state more consistently at positive voltages. Similarly, when KCl buffers were present, **TBP2** transported K^+ by forming channels that fluctuated rapidly between closed and open conformations at -80 mV. This rapid fluctuation implies dynamic gating mechanisms that respond differently to specific ion types and voltages. More interestingly, **TBP2** did not conduct ions in absence of voltage in KCl or NaCl buffer solution, indicating

no puncture in the lipid membrane. [Chui, J. K.; Fyles, T. M., Ionic conductance of synthetic channels: analysis, lessons, and recommendations. *Chem. Soc. Rev.* 41, 148-175 (2012)].

Figure 2J. Membrane colocalization of **TBP1** and **TBP2** in GUVs. Scale bars represent 5 μm (top row) and 2 μm (bottom row).

Figure 2. (H) Safranin O assay for membrane polarization in the presence of HEPES-NaCl (external) and HEPES-KCl (internal) buffers. (I) CF release assay; determination of CF release percentage in the presence of **TBP1** and **TBP2** (External: 10 mM HEPES 100 mM NaCl, pH 7.4; Internal: 10 mM HEPES 100 mM NaCl, pH 7.4) after 8 minutes.

The fluorescence microscopy images (Figure 2J) demonstrated that **TBP2** embedded within lipid membrane as indicated by the co-localization of **TBP2** with the membrane staining dye Nile red. This suggests that **TBP2** binds to the vesicular membrane. The CF release assay (Figure 2I, S2-SI) was further employed to investigate whether **TBPs** can disrupt the lipid membrane or vesicular structures. The data revealed that **TBP2** conserved the membrane integrity while **TBP1** formed larger openings, releasing the CF dye ($<10 \text{ \AA}$ in size) at significantly higher percentage ($\sim 52\%$) into the extravascular solution. The CF data correlates with the fluorescence microscopy data, indicating **TBP2** could possess the ability to form channels without rupturing the membrane. Besides, a safranin O assay was performed to investigate **TBP2**-dependent membrane polarization using unilamellar vesicles. Upon addition of safranin O to the vesicular solution, the fluorescence (with or without **TBP2**) (1 μM) was monitored for 600 seconds at an excitation of 522 nm and emission of 581 nm (Figure 2H). The data revealed a significant decrease in the fluorescence intensity of safranin O with increasing ligand concentration, indicating **TBP2**-induced membrane polarization in the presence of alternate NaCl or KCl buffers. These results demonstrate that **TBP2** embedded within the membrane and formed channel to facilitate ion transport via the vesicular system without rupturing the membrane.

3. I think the model was biased by the DFT optimization performed on the structure, which was more or less in a channel-like structure. A more convincing piece of evidence would be that a set of **TBP2** in

random positions in a lipid membrane is simulated to form a channel-like structure.

Transmission Electron Microscopy (TEM) imaging revealed that **TBP2** possesses the ability to self-organize into nanofibrous and vesicle like structures in both NaCl and KCl buffers. The dynamic light scattering (DLS) study further revealed the relative abundance of vesicles formed through the peptidomimetic **TBP2**'s supra-molecular assembly with an average size distribution of approximately 10 nm (Figure 1(ii)B). The formation of multiple vesicular structures was influenced by both the hydrophobic and hydrophilic parts of the compounds and strong aggregation of aromatic surfaces. However, in case of peptidomimetic **TBP1** with the Boc protecting group, no ordered structures similar to those of **TBP2** were observed in the presence of either Na⁺ or K⁺ ions (see the Supplementary Information, Figure S6). In the TEM image of **TBP1**, only a limited number of vesicular structures were observed, and they were not as distinct as those seen in **TBP2**. Consequently, the presence of the -NH₂ group in **TBP2** played a crucial role to promote the formation of well-organized non-covalent structures.

Figure R2. Snapshots at different time instants from the simulation of six self-assembled **TBP2** molecule in an aqueous solution containing 0.15 (M) NaCl.

The folded structure of **TBP2** as obtained from the DFT calculations, was also confirmed from MD simulations, where single **TBP2** molecules were observed to fold spontaneously in aqueous environment. Therefore, it is expected that, **TBP2** molecules self-assemble to form the cylindrical structure in the extracellular fluid, especially when there is a pool of **TBP2** molecules present in the immediate vicinity. To demonstrate the same, we have taken the DFT-optimized self-assembled and stacked structure of six **TBP2** molecules, and allowed to undergo MD simulation freely in an aqueous environment for 500 ns and Figure R2 represents snapshots at different time instants from this simulation. From **Figure R2**, it has been observed that, even after 200 ns, the self-assemble structure of **TBP2** was retained. Partial disruption of the structure started after 300 ns, and at the end of 500 ns, four **TBP2** molecules remained in their stacked conformation. Therefore, it is expected that the self-assembly of **TBP2** in an extracellular environment would produce patterned and stacked aggregates, and not an asymmetric structure consisting of randomly placed **TBP2** molecules. Also, spontaneous stacking of the folded **TBP2** molecules was observed in our MD simulations with free **TBP2** molecules in an aqueous environment. In the presence of lipid membranes, due to the packing of the adjacent lipid molecules, and their hydrophobic nature, the possibility of disruption of the **TBP2** aggregates is expected to be further reduced, bestowing higher stability to the ion-channel. Indeed, the same has been observed in our ion passage simulations, where unrestrained **TBP2** aggregates could spontaneously pass ions from one side of the lipid membrane to the other. This is also demonstrated by the video provided along with the supplementary information.

Additionally, the number of the stacked **TBP2** molecules was chosen in accordance with the thickness of the lipid bilayer, which represent a model system, however, the actual number of molecules stacked inside a lipid bilayer in experiments may vary, it might be higher or lesser as compared to that assumed in our MD simulations. Varying the number of the **TBP2** molecules in their aggregate may slightly change the average passage time of the ions, however, it does not change the fundamental mechanism. As shown in the video, ions passing through the channel comprising the stacked **TBP2** aggregates, move back and forth between consecutive **TBP2** molecules, and eventually gets released.

Simulation of randomly placed **TBP2** molecules inside a lipid membrane would necessarily involve interaction with lipid molecules, thereby delaying the process of aggregation, which in order makes it significantly difficult to simulate, and combined with the size of the overall system, is plagued by the requirement of immense computational time. Furthermore, the aggregation into an ion-channel also involves probability and initial velocity factors. Therefore, the reverse methodology was applied taking an aggregated structure of channel forming molecules, followed by placing them inside the lipid-pore. Similar strategies have been adopted in several MD simulation studies (e.g., *Chem. Rev.* **2020**, 120, 10298–10335; *Nat. Commun.* **2019**, 10, 2490, <https://doi.org/10.1038/s41467-019-10420-9>; *Angew. Chem. Int. Ed.* **2023**, 62, e202313712), depicting the passage of ions through ion-channels.

4. Some paragraphs do not refer their statements to specific figures, which are quite dense with many sub-figures. It causes some difficulties in understanding the data and how to relate it to the statements in the text.

The manuscript has been rigorously checked for such sentences.

5. The authors might refer to some DFT calculations of Na⁺ and K⁺ for their interactions with different environments, such as, DOI: 10.1021/jp510560k.

We have now compared the DFT calculated binding energies found in our study with those obtained from the above mentioned study. Also, the above reference has been included in the main manuscript.

6. Gramicidin A (gA) channel might be an interesting counterpart to compare the proposed model with (which I doubt it would be like as proposed). For gA, a typical barrier for K⁺ and Na⁺ 4-8 kcal/mol, and conductance is less than 24 pS (see 10.1021/acs.jctc.0c00968), and it is of course quite a stable channel.

We have compared the free energies of ion transport for both Na⁺ and K⁺, found in the present study with those observed in the case of gramicidin A (gA), citing the above-mentioned publication in the manuscript.

7. For studies of cell deaths, I am not convinced that **TBP2** would only kill cancer cells. The authors should present clear evidence for the action of **TBP2** for normal cells versus cancer cells. From the current evidence, I could argue that **TBP2** may be highly toxic. The argument that "**TBP2** possessed the ability to stabilize G4 structures in cancer cells", thus "induce cancer cell deaths" is highly speculative. There are many possibilities that a ligand can kill a cell. It may stop signaling pathways or may cause damages to mitochondria by simply blocking VDAC1 channels, or interfere with transcription processes. Killing a cell is easy, but killing only cancerous cells is actually a very hard problem. The author must demonstrate such a specificity to make such a claim.

The authors appreciate the reviewer's concern. The cytotoxic effect of **TBP1** and **TBP2** was assessed after a minimum of 24 h of treatment in both cancer and normal kidney epithelial (NKE) and Human Embryonic Kidney (HEK293T) cells. The cell viability study suggested that **TBP2** induced cancer cell death [IC₅₀ values of ~8.2, ~10 and ~16 μM in cervical carcinoma (HeLa), leukemia (K562) and lung carcinoma (A549) cells, respectively], while normal kidney epithelial (NKE) cells remained unaffected up to 100 μM. As per suggestion from the reviewer, we further examined the effect of **TBP2** in human embryonic kidney epithelial (HEK293T) cells in addition to NKE cells (Figure S12, Supplementary

Information). **TBP2** did not possess toxicity for HEK293T cells up to 100 μM as well. The low toxicity of thiazole peptide **TBP2** might be attributed to the less prevalence of G4 structures in normal cell lines like NKE or HEK293T. Additionally, the IC_{50} values of **TBP2** for K562 ($\sim 10 \mu\text{M}$) and A549 ($\sim 16 \mu\text{M}$) cells are considered to be moderately toxic, indicating its high toxicity towards HeLa ($\sim 8.2 \mu\text{M}$) cells. Thus, the data clearly demonstrated that **TBP2** exhibited toxicity (ranging from high to moderate) for cancer cell lines but showed no considerable toxicity for normal cell lines, as determined via the XTT-based cell viability assay.

Figure S12B. XTT based cell viability assay of **TBP2** in cancer (HeLa, A549 and K562) and normal (NKE and HEK293T) cell lines after 24 h of treatment.

However, we performed fluorescence experiments to examine **TBP2** induced changes in intracellular Na^+ or K^+ concentrations by using specific fluorescent dyes e.g., sodium-binding benzofuran isophthalate acetoxymethyl ester (SBFI-AM, Na^+ probe) and potassium-binding benzofuran isophthalate acetoxymethyl ester (PBFI-AM, K^+ probe) in HeLa, or A549 cells. HeLa or A549 cells were incubated with SBFI-AM or PBFI-AM ($10 \mu\text{M}$) for 2 h under standard culture conditions, followed by treatment with varying concentrations of **TBP2** (ranging from 0 to $8 \mu\text{M}$) for 1.5 h. The intracellular fluorescence measurements revealed that **TBP2** ($8 \mu\text{M}$) could significantly increase the concentration of Na^+ up to $\sim 37\%$ and marginally decrease the cytosolic K^+ concentration $\sim 15\%$ in HeLa cells (Figure 5E). However, $\sim 19\%$ increase in Na^+ and $\sim 10\%$ decrease in K^+ concentration were observed in **TBP2** embedded A549 cells compared to control. These results could be rationalized by different Na^+ or K^+ distributions across the cell membranes as the intracellular Na^+ concentrations ($\sim 12 \text{ mM}$) are much lower than the extracellular Na^+ ($\sim 145 \text{ mM}$), whereas the extracellular potassium concentration ($\sim 4 \text{ mM}$) are much lower than its intracellular counterpart ($\sim 150 \text{ mM}$) in physiological conditions. Thus, the data suggested that significant intracellular increase of monovalent cation Na^+ might disrupt the ionic balance. Being the basic structural component of G-quadruplex, Na^+ increase in the cellular system could synergistically contribute to the formation and widespread stabilization of G-quadruplex structures in cancer cells. This effect is likely facilitated by the substantial accumulation of monomeric **TBP2** molecules within the nuclei of cancer cells after an extended incubation period (Figure 5A).

Figure 5E. TBP2 dependent intracellular fluorescence intensity measurements of Sodium-binding benzofuran isophthalate acetoxymethyl ester (SBFI-AM, Na⁺ probe) and Potassium-binding benzofuran isophthalate acetoxymethyl ester (PBFI-AM, K⁺ probe) in i) HeLa and ii) A549 cells. iii) Schematic illustration of TBP2-G4 interaction.

Figure 5A. Confocal images of HeLa cells (fixed) stained with TBP2 (blue) and BG4 (red); scale bars represent 10 μm . Nucleus of each cell is indicated with white borderline.

We have performed the fluorescence based confocal microscopy to illustrate the cell permeability and G4 stabilization property in HeLa cells. As observed in Figure 5A, TBP2 (blue) efficiently penetrates the cell membrane and accumulates inside the HeLa cell nuclei (nucleus is indicated with white line). The images also suggested that the concentration of TBP2 was significantly higher inside the cellular nuclei

compared to cytoplasm of HeLa cells. The number of G4 binding antibody (BG4) foci (as seen as punctuated dots) was significantly increased in **TBP2** treated cells compared to the control cells or **TBP1** treated cells (Figure 5A, B), suggesting the potential of **TBP2** to stabilize G4s in cellular system.

8. I feel that the manuscript has two separate stories, one is to establish a channel-like structure of **TBP2** assemblies, while the other is the interaction of **TBP2** and G4 and its implication for treating cancer. Both stories are quite interesting and warrant publications somewhere else, even though the writing of this MS makes it hard to follow and understandable to a wider audience.

Scheme 1. Diagrammatic representation of **TBP2** gated ion channel formation and cell death in time-dependent manner.

-In the revised manuscript, we have shown that **TBP2** rapidly localizes in model lipid bilayers and the plasma membrane to facilitate ion transport, while it accumulates in the nuclei of HeLa cells after prolonged duration (e.g., 24 h) in a time-dependent manner. The formation of channels by **TBP2** leads to increase in intracellular concentration of monovalent cations like Na⁺, the essential components of DNA G4 structures. The interplay of time-dependent membrane or nuclear localization of **TBP2**, followed by its increased intracellular cation concentration and direct interaction with G4, synergistically promotes G4 formation and stabilization, triggering cancer cell death.

Reviewer #1 (Remarks to the Author):

The authors have sincerely addressed the concerns raised by this reviewer and have made appropriate revisions to the manuscript. I now recommend that this manuscript be published in Nature Communications.

Reviewer #3 (Remarks to the Author):

In response to my first point in the rebuttal letter, the author did not address why they chose 6 TBP2 while the hill coefficient is only 1.64. Later on, in their MD simulation of 500 ns, their initial DFT assembly clearly has few TBP2 dissociated from a stack of 6 TBP2, significantly re-arranged so that it is no longer a channel-like structure as initially hypothesized. In my experiences, within 500 ns the structure deforms so much like that, it is unlikely to sustain its channel-like structure if running longer, perhaps within few microseconds, which would only take few days nowadays. This clearly counters the hypothesis that a channel-like structure of 6 TBP2 could be formed in lipid membrane, of which the authors appear reluctant to perform a simulation.

For the response related to the current data (i.e., Figure R1), I am still not convinced this is a channel-like structure. The reason I brought up Gramicidin A (gA) was that this channel is formed by two identical peptides, each of which has 15 residues. If the authors look at the current trace of this gA channel, it has a step-wise behavior as a function of time, and yet there is no gating mechanism known for this channel. The different step-wise currents of gA were attributed to different conducting modes, while the zero current would indicate that there is no channel formed by the two identical peptides during a period of time. So, the near zero-value current in Figure R1 may suggest either no formation of a possible conducting pore just like gA, or some leaking currents may occur. Note that leaking currents do occur in ion channels. Some known leaking currents happen through voltage sensor domains away from the conducting pore of a channel (e.g., hERG channel). This suggests that the existence of some leaking currents should not be attributed to the existence of an actual channel-like structure.

Now suppose if a large conducting pore is formed, obviously there is an ion flow. But for a hypothesized channel of TBP2, such an ion flow is not steady like observed in an actual ion channel. Indeed, If the authors look closely to the current trace for KCl, some durations have the current slowly increases, then suddenly disappears. This is not channel-like behavior at all. This does suggest some fast dynamical re-arrangement of TBP2 assembly in the lipid membrane as seen in the MD simulations (Figure R2). But claiming that a channel structure is formed, I think, is not supported.

The argument that "TBP2 did not conduct ions in absence of voltage in KCl or NaCl buffer solution, indicating no puncture in the lipid membrane" is not really valid because the authors must rule out any possible punctures or non-channel-like structures WHEN TBP2 assembly conducts ions to prove the hypothesis.

To this end, even though the authors did their best to address the issues, I still have a lot of doubts about the conclusion drawn by the authors. I think a large audience of Nature would likely have similar doubts about the conclusion as well.

Point-by-point response to Referees

REVIEWER COMMENTS

Reviewer #1 (Remarks to the Author):

The authors have sincerely addressed the concerns raised by this reviewer and have made appropriate revisions to the manuscript. I now recommend that this manuscript be published in Nature Communications.

The authors thank the reviewer for appreciating the revised manuscript and accepting the manuscript in Nature Communications.

Reviewer #3 (Remarks to the Author):

In response to my first point in the rebuttal letter, the author did not address why they chose 6 **TBP2** while the hill coefficient is only 1.64.

- We thank the reviewer for the concerns regarding the chosen number of **TBP2** molecules within the lipid membrane for the formation of the ion channel. As mentioned previously, the Hill equation was used to determine the EC₅₀ values (concentration of a molecule to achieve 50% ion transport activity) of the peptidomimetics. **TBP2** exhibited EC₅₀ values of 5.8 μM and 7 μM for K⁺ and Na⁺, respectively. The Hill coefficient (n) assumes the possible co-operativity of a ligand to act as channels through the lipid membrane. A recent study (Talukdar et. al, *Angew Chem*, e202319919, 2024) demonstrates that the determined Hill coefficient (n) values for ligands **3a** (n = 2.2), **2a** (EC50 = 0.58 μM and, n = 2.0), and **1a** (EC50 = 3.98 μM and n = 2.41) indicates the supramolecular dimer formed by monomers of **1a** –**3a** as the active rosette structure for the supramolecular assembly of ion channels. In our study, the Hill coefficient for K⁺ (nK⁺ = 1.64) indicates the positive co-operativity of **TBP2** to form supramolecular channel like structures to transport K⁺. Besides that, the Hill coefficient value >1 but <2, suggest that one **TBP2** molecule or the monomer participates to form the rosette like structure for the supramolecular nanochannel assembly which corroborates with the quantum chemistry calculations and DFT optimization [Gilles, A.; Barboiu, M., Highly selective artificial K⁺ channels: an example of selectivity-induced transmembrane potential. *J. Am. Chem. Soc.* 138, 426-432 (2016); Malla, J. A. et al. A glutathione activatable ion channel induces apoptosis in cancer cells by depleting intracellular glutathione levels. *Angew Chem* 59, 7944-7952 (2020); Sharma, R., Sarkar, S., Chattopadhyay, S., Mondal, J. & Talukdar, P. A Halogen-Bond-Driven Artificial Chloride-Selective Channel Constructed from 5-Iodoisophthalamide-based Molecules. *Angew Chem*, e202319919 (2024)].

Furthermore, the supramolecular structure of **TBP2** was determined by optimizing the molecule using quantum chemistry calculations at different levels (M06-2x/6-31G(d,p) and B3LYP/6-31G(d,p)). The optimized structure revealed that **TBP2** became folded and stabilized by three hydrogen bonding interactions, one of which involves the terminal -NH₂ group (Figure 1(ii)D and 1(ii)E). The hydrogen bonding distances were calculated to be 1.85 Å, 1.88 Å, and 2.31 Å, respectively. The structure exhibited an almost elliptical cavity, with dimensions of approximately 5.8 Å for the long axis and 4.6 Å for the short axis. Subsequently, Na⁺/K⁺ ions were placed within this cavity, and their coordination with the molecular structure was investigated through DFT optimization (Figure 1(ii)F and 1(ii)G). The cations were observed to be well-contained within the cavity without causing significant changes to the surrounding molecular arrangement. The cations were coordinated to three donor atoms, a triazole nitrogen, a nitrogen atom from a thiazole ring and a carbonyl oxygen from a peptide linkage, and the corresponding distances are shown in Figure 1(ii) F, and 1(ii) G. To quantify ion encapsulation, the binding energies of Na⁺ and K⁺ were calculated using the formula $E_{\text{binding}} = E_{\text{(M-ion)}} - E_{\text{M}} - E_{\text{ion}}$, where E_(M-ion), E_M, and E_{ion} represent the electronic energies of the optimized structures of the molecule-ion composite system, the free molecule, and the cation, respectively. The binding energies for Na⁺ and K⁺ were calculated to be -75.6 kcal/mol and -55.9 kcal/mol, respectively at the M06-2x/6-

31G(d,p), and -74.6 kcal/mol, and -67.7 kcal/mol, respectively at B3LYP/6-31G(d,p) level. The high binding energies indicate the strong stability of the folded molecular structure to accommodate Na^+ and K^+ ions within its cavity, and are similar to those found for these metal ions bound to the active sites of various proteins.

In the previous reply to the reviewer, it was stated that the number of the stacked **TBP2** molecules was chosen in accordance with the thickness of the lipid bilayer, which represent a model system, however, the actual number of molecules stacked inside a lipid bilayer in experiments may vary, it might be higher or lesser as compared to that assumed in our MD simulations. Varying the number of the **TBP2** molecules in their aggregate may slightly change the average passage time of the ions, however, it does not change the fundamental mechanism. As shown in the movie, ions passing through the channel comprising the stacked **TBP2** aggregates, move back and forth between consecutive **TBP2** molecules, and eventually gets released. Therefore, the same mechanism would be followed even if greater number of molecules takes part in the formation of the ion channel.

To further demonstrate why we have chosen the number of **TBP2** molecules to be like that they have nearly equal thickness compared to the lipid membrane, we have now kept an optimized structure of two **TBP2** molecules inside a membrane pore and performed simulations for 200 ns. It is observed that, immediately after start of the simulation, the pore collapses and the ion channel architecture is lost. It is expected since the lower intermolecular interaction and the height of the assembly of two **TBP2** molecules cannot be stabilized inside a pore of lipid which has a length equal to the height of the lipid membrane. Several snapshots at various time instants from this simulation are provided below (Figure R1). It has been observed that at the end of 100 ns, **TBP2** molecules are expelled out of the inside of the membrane and they travel to the surface of the same. Therefore, in order to maintain the hydrophobic interactions between the lipid tails, the pore collapses and the **TBP2** molecules are transferred to the more hydrophilic domain of the membrane. Therefore, clearly more than 2 molecules are required for stabilization of the lipid pore to conduct ions.

Figure R1. Snapshots showing side and top views of a lipid membrane with a pore containing an assembly of two **TBP2** molecules at different time instants. The initially created pore starts to collapse immediately after starting of the simulation and the collapse is complete within 50 ns of simulation followed by travelling of the aggregate of the **TBP2** molecules from the lipid core to the lipid surface.

Later on, in their MD simulation of 500 ns, their initial DFT assembly clearly has few **TBP2** dissociated from a stack of 6 **TBP2**, significantly re-arranged so that it is no longer a channel-like structure as initially hypothesized. In my experiences, within 500 ns the structure deforms so much like that, it is unlikely to sustain its channel-like structure if running longer, perhaps within few microseconds, which would only take few days nowadays. This clearly counters the hypothesis that a channel-like structure of 6 **TBP2** could be formed in lipid membrane, of which the authors appear reluctant to perform a simulation.

To show the stability of the aggregate of the **TBP2** molecules, in the previous communication, we performed free simulations of the **TBP2** assembly of 6 molecules in water, where it is certainly expected to dissociate, due to entropy effects. However, inside the lipid membrane, due to the packing of the adjacent lipid molecules, and their hydrophobic nature; the possibility of disruption of the **TBP2** aggregates is not expected. As suggested by the reviewer, we have now conducted 2500 ns (2.5 μ s) long simulations with the **TBP2** assembly of 6 molecules, placed inside a lipid pore. As observed from Figure S9(a), the structure of the stacked arrangement of the six **TBP2** molecules is maintained throughout the entire duration of the trajectory without the application of any restraining force unlike the case of two **TBP2** molecules as observed above. Also, from the visual inspection of Figures S9(b) and S9(c), it is evident that the ion channel is maintained even after 2.5 μ s, and no **TBP2** molecule was expelled out of the membrane. The stability of the ion channel was also reflected in the average heavy-atom RMSD which fluctuated between 0.25 and 0.38 nm, and the average intermolecular distance between the **TBP2** molecules, which did not change appreciably, the difference between the maximum and minimum distance being only \sim 0.15 nm, which is indeed insignificant. Minor reorganizations of the **TBP2** assembly took place due to the structural pressure of the lipid molecules in the immediate vicinity of the pore as observed from the fluctuation of the RMSD and intermolecular distance around 1400 ns and 2000 ns, however, the **TBP2** channel was clearly maintained, and the final structure was grossly similar to the initial configuration. These findings imply that we might rule out the possibility of severe damage of the **TBP2** assembly and change in the aggregation pattern inside the lipid pore even at a high time scale. Additionally, it is evident that the number of **TBP2** molecules would be such that they span the entire thickness of the lipid membrane, otherwise it is unlikely that the ion channel would be maintained without being engulfed by the membrane itself. Let us hypothesize a set of four **TBP2** molecules inside a lipid pore. Even if we consider that they get stabilized inside the lipid pore without distorting the structure of the ion channel, there would be vacant space in the membrane above and below the molecules, where rearrangement of lipid molecules may happen, as expected from the self-interaction pattern of usual lipid molecules in a membrane, which could close the lipid pore, making the ion transport nearly impossible.

Figure S9. (a) Snapshots representing structures of the six **TBP2** molecules embedded in the lipid pore at different time instants spanning 2500 ns of production simulation time. (b) Top, and (c) side views of the initial (0 ns) and final (2500 ns) structures of the lipid membrane containing the ion channel. (d) Root mean squared displacement (RMSD) of the heavy atoms for the **TBP2** molecules, and (e) the average distance between consecutive **TBP2** molecules throughout the trajectory.

For the response related to the current data (i.e., Figure R1), I am still not convinced this is a channel-like structure. The reason I brought up Gramicidin A (gA) was that this channel is formed by two identical peptides, each of which has 15 residues. If the authors look at the current trace of this gA channel, it has a step-wise behavior as a function of time, and yet there is no gating mechanism known for this channel. The different step-wise currents of gA were attributed to different conducting modes, while the zero current would indicate that there is no channel formed by the two identical peptides during a period of time. So, the near zero-value current in Figure R1 may suggest either no formation of a possible conducting pore just like gA, or some leaking currents may occur. Note that leaking currents do occur in ion channels. Some known leaking currents happen through voltage sensor domains away from the conducting pore of a channel (e.g., hERG channel). This suggests that the existence of some leaking currents should not be attributed to the existence of an actual channel-like structure. Now suppose if a large conducting pore is formed, obviously there is an ion flow. But for a hypothesized channel of **TBP2**, such an ion flow is not steady like observed in an actual ion channel. Indeed, If the authors look closely to the current trace for KCl, some durations have the current slowly increases, then suddenly disappears. This is not channel-like behavior at all. This does suggest some fast dynamical re-arrangement of **TBP2** assembly in the lipid membrane as seen in the MD simulations (Figure R2). But claiming that a channel structure is formed, I think, is not supported.

The authors thank the reviewer for the comment. We agree with the reviewer that gramicidin A possesses different conducting modes (step-wise behavior) which are consistent throughout. In our case, we observed the current trace in the presence of different ions like Na⁺, K⁺, Rb⁺, Cs⁺, Li⁺. The current data suggest that **TBP2** demonstrates step like behavior in the presence of Na⁺ while it shows square top behavior in the presence of K⁺, indicating its responsiveness to different ions in relation to applied voltages (Figure 3). **TBP2** displayed multiple channel openings at -80 mV but shows single channel behavior at the applied potential of +80 mV to conduct Na⁺. **TBP2** possesses single channel behavior at the holding potential of +80 mV in the presence of K⁺ as well. Besides that, it does not show identical channel forming activities in the presence of Rb⁺, Cs⁺, Li⁺ (Figure S4, Supplementary Information).

Figure S4. Current Vs. voltage data of **TBP2** at the applied potential of +100 mV in the presence of 1 M KCl for the duration of 15 seconds.

We also acknowledge the comment of the reviewer: *Indeed, If the authors look closely to the current trace for KCl, some durations have the current slowly increases, then suddenly disappears. This is not channel-like behavior at all.* In the manuscript, Figure 1D represents a current trace of 1 second in 1 M KCl at +80 mV. Here, we have given a long trace of **TBP2** mediated current via the lipid bilayer up to 15 seconds in the presence of 1 M KCl at the applied voltage of +100 mV (Figure S4, Supplementary Information). The data clearly suggests that **TBP2** holds the single channel activity for a longer duration, indicating its potential to form channel via the lipid membrane.

Apart from that, the authors acknowledge the reviewer's concern regarding the conclusiveness of **TBP2** molecules forming channels in the membrane. The ion channel experiments are primarily based on the fluorescence based assays (e.g., HPTS assay, CF release assay, Cl⁻ specific lucigenin assay, safranin O based membrane polarization assay) and electrophysiology experiments (e.g., patch clamp) in synthetic lipid bilayer systems [Gilles, A.; Barboiu, M., Highly selective artificial K⁺ channels: an example of selectivity-induced transmembrane potential. *J. Am. Chem. Soc.* 138, 426-432 (2016); Debnath, M.; Chakraborty, S.; Kumar, Y. P.; Chaudhuri, R.; Jana, B.; Dash, J., Ionophore constructed from non-covalent assembly of a G-quadruplex and liponucleoside transports K⁺-ion across biological membranes. *Nat. commun.* 11, 1-12 (2020); Montenegro, J.; Ghadiri, M. R.; Granja, J. R., Ion channel models based on self-assembling cyclic peptide nanotubes. *Acc. Chem. Res.* 46, 2955-2965 (2013)]. In the manuscript, we present a set of experimental data and analyses that collectively support that **TBP2** indeed functions as an ion channel.

Fluorescence Microscopy and Co-localization: The fluorescence microscopy images (Figure 2J) demonstrated that **TBP2** embedded within lipid membrane as indicated by the co-localization of **TBP2** with the membrane staining dye Nile Red. This suggests that **TBP2** binds to the vesicular membrane.

CF Release Assay: The CF release assay (Figure 2I, S2-SI) was employed to investigate whether **TBPs** can disrupt the lipid membrane. The data revealed that **TBP2** conserved membrane integrity while **TBP1** formed larger openings, releasing the CF dye (<10 Å in size) at significantly higher percentage (~52%) into the extravesicular solution. The CF data correlates with the fluorescence microscopy data, indicating **TBP2** could possess the ability to form channels.

HPTS Assay: We utilized HPTS (pH-sensitive fluorescent dye) assays (Figure 2A-F) to show that **TBP2** could potentially transport metal ions such as Na⁺ and K⁺. The increase in HPTS fluorescence intensity indicated ion influx, and **TBP2's** EC₅₀ values for K⁺ and Na⁺ further confirmed its ion transport activity. The fluorescence intensity of HPTS increased due to formation of channels leading to H⁺ efflux following influx of cations e.g., Na⁺, K⁺, Rb⁺, Cs⁺, Li⁺ etc. **TBP2** showed the highest ion transport activity across the model lipid bilayer at 20 μM concentration for K⁺ (~ 70%) and Na⁺ (~ 65%) in HEPES-NaCl or HEPES-KCl buffer (pH 7.4). The Hill equation was used to determine the EC₅₀ values (concentration of

molecules to achieve 50% ion transport activity) of the peptidomimetics. **TBP2** exhibited EC₅₀ values of 5.8 μM and 7 μM for K⁺ and Na⁺, respectively. The Hill equation used here is as follows:

$$I = 1/(1 + (EC_{50}/[Channel])^n)$$

I = relative fluorescence intensity, [Channel] = concentration of **TBP1** or **TBP2**, n = Hill co-efficient. The EC₅₀ values of **TBP2** for K⁺ and Na⁺ were determined to be 3.1 μM and 14 μM while **TBP1** showed EC₅₀ values of 43 μM and 56 μM for K⁺, and Na⁺, respectively (Internal buffer: HEPES NaCl, pH 6.4) (Figure 2 and see the SI, Figure S1). When the internal buffer was substituted with KCl (10 mM HEPES 100 mM KCl, pH 6.4), **TBP2** displayed EC₅₀ values of 5.8 μM and 7 μM for K⁺ and Na⁺, respectively. Thus, HPTS studies revealed that **TBP2** can form channels, enabling efficient transportation of ions while **TBP1** did not show channel-forming activities.

Lucigenin Assay: In the revised manuscript, we employed a lucigenin assay (Figure 2G) to observe Cl⁻ transport activity of **TBP2**. The fluorescence intensity of lucigenin did not change significantly compared to blank with incremental addition of **TBP2**; suggesting no Cl⁻ flux across the lipid membrane.

Safranin O Assay: The safranin O assay (Figure 2H) was also conducted to investigate **TBP2**-dependent membrane polarization using GUVs. The data revealed that **TBP2** modulates membrane polarization in the presence of NaCl and KCl buffer solutions. For membrane polarization experiments, GUVs were prepared using 9:1 mixture of 10 mM EYPC and cholesterol via electroformation technique. In a fluorescence cuvette, 25 μL of GUV solution was suspended in 475 μL 10 mM HEPES, pH 6.4 containing the corresponding salt 100 mM NaCl or KCl. Next, safranin O was added at a final concentration of 1 μM. The fluorescence (with or without **TBP2**) was monitored for 600 sec at an excitation of 522 nm and emission of 581 nm (Figure 2H). The data showed that fluorescence intensity of safranin O significantly decreased with increasing concentration of ligand, suggesting **TBP2** modulated membrane polarization in the presence of NaCl (external) and KCl (internal) buffer solution.

Figure 2. G) Lucigenin assay of **TBP2** for Cl⁻ transport. Change in fluorescence intensity as a function of time in 225 mM NaNO₃ buffer. (H) Safranin O assay for membrane polarization in the presence of HEPES-NaCl and KCl buffers. Change in fluorescence of safranin O as a function of time (with or without **TBP2**).

Patch Clamp Experiments: We performed patch clamp experiments (Figure 3) using planar lipid bilayers, which show distinctive channel openings and closings upon addition of **TBP2**, confirming its channel-forming behavior for both Na⁺ and K⁺.

Patch clamp experiments using planar lipid bilayers provided further insight into real-time channel formation behaviour of thiazolyl peptidomimetic **TBP2** (Figure 3). The addition of **TBP2** to the planar bilayer resulted in distinctive channel openings and closings at +80 mV and -80 mV, confirming the formation of channels across planar lipid bilayer membrane in the presence of both Na⁺ and K⁺ (Figure 3). I-V plot of **TBP2** shows an ohmic-linear relationship between current vs. voltage (Figure 3G). In the presence of 1 M NaCl at both *cis* and *trans* side, **TBP2** displayed multiple channel openings

to transport Na^+ at -80 mV (Figure 3A). It also exhibited multiple square-top behavior at the holding potential of +80 mV in the presence of either NaCl or KCl. It formed multiple channel openings that were stable for longer duration (~2 sec) in the presence of Na^+ whereas it transported K^+ by forming channels that fluctuates more quickly between the closed and open conformations (Figure 3). **TBP2** efficiently transported Na^+ and K^+ across planar lipid bilayer membrane with high conductance. The average conductance values for transporting Na^+ and K^+ were measured to be ~0.56 nS and ~0.68 nS in the presence of 1 M NaCl and KCl, respectively. However, the current behaviour was also examined in the presence of CsCl, LiCl, RbCl buffers to illustrate the voltage dependent gating characteristics of **TBP2** for other monovalent cations (e.g., Cs^+ , Li^+ , Rb^+) besides Na^+ or K^+ . I-V curve suggests that **TBP2** possessed comparatively lower conductance values for other metal ions (e.g., Cs^+ : 0.29 nS, Li^+ : 0.08, Rb^+ : 0.15 nS) than Na^+ or K^+ in relation to applied positive or negative voltages (Figure 3).

Molecular Dynamics Simulation: First of all, we want to mention that, we did not find any significant structural rearrangement of **TBP2** molecules within the lipid membrane creating damage to the ion channel. In all of our simulations regarding the ion channel present within the lipid pore, the channel was observed to be intact throughout the simulation trajectories, the maximum of which was continued up to 2500 ns. We only observed partial dissociation of **TBP2** molecules under free state in water in absence of membrane, which is expected due to entropy effects. Molecular dynamics simulations were employed to study the supramolecular arrangement of **TBP2** and its mechanism of ion channel formation. The simulation results suggested that **TBP2** can spontaneously transport Na^+ and K^+ ions through the channel, further supporting its ionophoric capabilities.

The molecular dynamics simulation illustrated the supramolecular arrangement of **TBP2** and provided mechanistic insights of ion channel formation in the lipid membrane. By performing molecular dynamics study, we tracked the number of Na^+ , K^+ or Cl^- present within the ion channel. Initially, no cations or anions were present within the channel. However, as shown in Figure 4, after only few picoseconds of simulations, cations entered the channel, and the number of cations continuously varied throughout the entire duration. We observed one or two Na^+ and K^+ ions to be present inside the channel for most of the time, while three cations were present in fewer time instants. An incessant change in the number of cations in the ion channel validated the exit of old ions as well as the entry of new cations, thereby suggesting spontaneous ion passage through the channel.

The average interaction energy of a single cation with the whole ion channel was computed to be -23.8 kcal/mol, and -14.7 kcal/mol for Na^+ and K^+ , respectively; which followed the same trend as obtained from our quantum chemistry calculations. Clearly, a Na^+ was more stable compared to a potassium ion within the channel.

In coherence with the DFT results, we observed only one chloride ion to be present within the stacked **TBP2** molecules in few time instants while in most of the time instants, no chloride ion entered the cavity (Figure 4C). Therefore, certainly the stacked **TBP2** molecular cavity had an inclination toward the binding and passage of cations of suitable size. Altogether, the experimental and molecular dynamics simulation study suggest that **TBP2** could act as ionophore to transport ions across lipid membrane.

Figure 4. (C) Number of ions within the channel for the 500 ns production simulations for the channel-embedded lipid membrane in 0.15 M NaCl and KCl medium. (D) Free energy profiles (in kcal/mol) in terms of potential of mean forces (PMF's) for the passage of Na⁺, K⁺, and Cl⁻ ions through the ion-channel at 310 K, calculated employing the adaptive biasing force (ABF) module implemented in NAMD 2.12.

The argument that “**TBP2** did not conduct ions in absence of voltage in KCl or NaCl buffer solution, indicating no puncture in the lipid membrane” is not really valid because the authors must rule out any possible punctures or non-channel-like structures WHEN **TBP2** assembly conducts ions to prove the hypothesis.

We thank the reviewer for the comment. Before proceeding into patch clamp experiments to measure the ion conductance, we first performed the fluorescence based imaging experiments to demonstrate the membrane embedding feature of **TBP1** and **TBP2** using a membrane staining dye Nile red. The fluorescence microscopy images (Figure 2J) suggest that **TBP2** embedded within lipid membrane as indicated by the co-localization of **TBP1** and **TBP2** with the membrane staining dye Nile red, illustrating the potential of **TBP2** to bind to the vesicular membrane. Next, the 5(6)-carboxyfluorescein (CF) release assay was conducted to observe any possible puncture or large pore in the membrane. In principle, a CF leakage assay was conducted using a self-quenched carboxyfluorescein dye (CF) (<10 Å in size) (Figure 2I, see Supplementary Information, Figure S2). The membrane-impermeable CF dye (40 mM) can efflux from the vesicles upon pore formation (>10 Å) or disruption of the LUVs, leading to an increase in fluorescence intensity. After addition of **TBP1** and **TBP2**, the CF discharge percentage was calculated as 52% and 2.7% after 8 minutes, respectively. The lower CF release in the presence of **TBP2** indicated conserved membrane integrity or no puncture via the LUVs, while the higher CF discharge suggested that **TBP1** could disrupt the membrane structure or create larger openings in the vesicles. Thus, the CF release data demonstrate that **TBP2** does not create puncture or disruption of the membrane. In addition, safranin O assay for membrane polarization experiment indicates **TBP2**-induced membrane polarization in the presence of alternate NaCl or KCl buffers. Therefore, microscopic images and fluorescence based assays confirm no puncture or disruption of the lipid membrane of large or giant unilamellar vesicles (LUVs or GUVs) by the thiazole based peptidomimetic **TBP2**. Besides, no significant currents were observed via the lipid membrane of GUVs in the presence **TBP2** with no applied potential.

Figure 2. (I) CF release assay; determination of CF release percentage in the presence of **TBP1** and **TBP2** (External: 10 mM HEPES 100 mM NaCl, pH 7.4; Internal: 10 mM HEPES 100 mM NaCl, pH 7.4) after 8 minutes.

Reviewer #3 (Remarks to the Author):

I am pleased to see the effort that tries to "falsify" some of the claims in the initial manuscript.

Even though I think 2.5 microseconds of the aggregate in a lipid membrane are too short given a dynamical behavior of the peptides, this is a good scientific way to examine some of the biases I can see in the hypothesis.

I would suggest the authors to briefly discuss some of the aspects in the Discussion that (1) simulations are not long enough to see large deviations from the initial structures and (2) effects of lipid membranes, which might be relevant for a separate study.

Overall, this is a quite interesting study, which might be worth further investigations.

Point-by-point response to referees

REVIEWER COMMENTS

Reviewer #3 (Remarks to the Author):

I am pleased to see the effort that tries to "falsify" some of the claims in the initial manuscript. Even though I think 2.5 microseconds of the aggregate in a lipid membrane are too short given a dynamical behaviour of the peptides, this is a good scientific way to examine some of the biases I can see in the hypothesis. I would suggest the authors to briefly discuss some of the aspects in the Discussion that (1) simulations are not long enough to see large deviations from the initial structures and (2) effects of lipid membranes, which might be relevant for a separate study.

Overall, this is a quite interesting study, which might be worth further investigations.

We thank the reviewer for carefully evaluating the merit of the revised manuscript. During our 2.5 microsecond long simulations, we did not observe any appreciable change in the structure of the **TBP2** assembly, while the lipid membrane maintained its structural integrity. Although the time scales of the MD simulations performed are smaller than the experimental time frame, those are significant as far as the contemporary standards in MD simulations with classical force-fields. Coarse-grained MD simulations could be a possible alternative methodology, although they may compromise the accuracy of simulation results to a considerable extent. In the case of two **TBP2** molecules embedded onto the lipid membrane, we observed expulsion from the membrane core within few nanoseconds only, however, for six **TBP2** molecules, the structure of the supramolecular assembly inside the lipid membrane remained grossly unaltered throughout the simulation (2500 ns). This observation suggests that the steric pressure of the lipid molecules do not allow significant structural reorganization of the assembly. Thus, we deduce that the structure of the supramolecular assembly would remain largely unchanged even with further continuation of MD simulations.

Furthermore, we did not observe any specific affinity of **TBP2** towards the lipid functional groups other than the inherent polar nature. In cases, where various lipid molecules are present, the fundamental nature of interactions remain the same. Since, the **TBP2** assembly is initially confined within the membrane and stabilized by insignificant self-reorganization, and lipid movement, the functional groups of the lipid membranes may interact with the **TBP2** assembly, thereby influencing the overall stability of the assembly inside the lipid pore. While there may be some changes in lipid ordering in the immediate vicinity of the **TBP2** assembly, we do not anticipate significant long-range structural disorder in the lipid membrane due to its massive size. However, further investigation is required to substantiate the hypothesis, which could be explored in future studies.